## PROCEEDINGS A

computational mechanics, mechanics, applied mathematics

gyroscopic system, friction-induced instability, slow–fast systems, homoclinic/heteroclinic connection, dynamics, global bifurcation

**Author for correspondence:**
Simon Sailer
e-mail: simon.sailer@inm.uni-stuttgart.de

# Singularly perturbed dynamics of the tippedisk

Simon Sailer and Remco I. Leine

Institute for Nonlinear Mechanics, University of Stuttgart, Pfaffenwaldring 9, 70569 Stuttgart, Germany

SS, 0000-0002-1275-9634; RIL, 0000-0001-9859-7519

The *tippedisk* is a mathematical-mechanical archetype for a peculiar friction-induced instability phenomenon leading to the inversion of an unbalanced spinning disc, being reminiscent of (but different from) the well-known inversion of the tippetop. A reduced model of the tippedisk, in the form of a three-dimensional ordinary differential equation, has been derived recently, followed by a preliminary local stability analysis of stationary spinning solutions. In the current paper, a global analysis of the reduced system is pursued using the framework of singular perturbation theory. It is shown how the presence of friction leads to slow–fast dynamics and the creation of a two-dimensional slow manifold. Furthermore, it is revealed that a bifurcation scenario involving a homoclinic bifurcation and a Hopf bifurcation leads to an explanation of the inversion phenomenon. In particular, a closed-form condition for the critical spinning speed for the inversion phenomenon is derived. Hence, the tippedisk forms an excellent mathematical-mechanical problem for the analysis of global bifurcations in singularly perturbed dynamics.

## 1. Introduction

The aim of the present paper is to perform a global analysis of the tippedisk, a spinning unbalanced disc in frictional contact with a support, by exploiting its singularly perturbed structure. The tippedisk forms a new mechanical-mathematical archetype that exhibits a friction-induced homoclinic bifurcation followed by a Hopf bifurcation, which explains the inversion phenomenon.

**Figure 1.** Inversion of the tippedisk, showing the rise of the COG (black dot).

Although the goal of nonlinear dynamics is to understand and predict nonlinear dynamic phenomena in engineering applications, it proves notoriously difficult to apply the body of methods and concepts provided by nonlinear dynamics to real-world applications. Several reasons for this can be named. First of all, a closed-form analysis of a nonlinear system can only be performed for a system with a few degrees of freedom, whereas models used in industry easily involve thousands of degrees of freedom. Furthermore, the concepts and fundamental theorems of nonlinear dynamics have been developed for ordinary differential equations (ODEs) with enough differentiability properties. The extension of these concepts to non-smooth systems, stochastic systems, delay differential equations, differential algebraic systems, partial differential equations and the like is still a topic of intense ongoing research. For this reason, one often finds that methods and concepts of nonlinear dynamics are explained, developed and tested on a set of ODEs that have virtually no resemblance to any real-world application. One may argue that nonlinear dynamics, as a branch in applied mathematics, can universally be applied and it therefore also suffices to use abstract models. However, by restricting the use of global analysis techniques (e.g. Melnikov theory) to either abstract ODEs or almost trivial systems (e.g. the pendulum equation) one risks to oversee the original goal of nonlinear dynamics. This motivates the quest for a set of easily understandable, non-trivial, 'real' problems on which global analysis techniques of nonlinear dynamics may be applied, and, at the same time, may be tested in a laboratory set-up. At this point, a number of gyroscopic 'scientific toy' systems enter the scene, which all consist of a single rigid body in frictional contact with a supporting hyperplane such as the Euler disc [1–3], the rattleback [4,5], spinning axisymmetric bodies [6–9] (e.g. spinning eggs [10]) and the tippetop [11–14]. Together, they form a mathematical playground to explain, develop and test novel methods in nonlinear dynamics without losing touch with the real world. This special feature of such types of systems explains that the research on the tippetop, which originated in the 1950s, is a topic of increased current research [15–17].

In [18], we introduced a new mechanical-mathematical archetype, called the tippedisk, to the scientific playground and derived a suitable mechanical model. Essentially, the tippedisk is an eccentric disc, for which the centre of gravity (COG) does not coincide with the geometric centre of the disc. Neglecting spinning friction (i.e. pivoting friction), two stationary motions can be distinguished. For 'non-inverted spinning', the COG is located below the geometric centre and the disc is spinning with a constant velocity about the in-plane axis through the COG and the geometric centre. The second stationary motion is referred to as 'inverted spinning', being similar to 'non-inverted spinning', but with the COG located above the geometric centre of the disc, see figure 1. If the non-inverted tippedisk is spun fast around an in-plane axis, the COG rises until the disc ends in an inverted configuration, shown in figure 2.

In [19], a reduced model of the tippedisk has been developed by making use of physical constraints and simplifying assumptions of the model derived in [18]. This three-dimensional model was preliminary studied by a linear stability analysis using Lyapunov's indirect method. Moreover, a closed-form expression has been derived, which characterizes the critical spinning velocity $\Omega_{\mathrm{crit}}$ at which a Hopf bifurcation occurs, indicating for supercritical spinning velocities

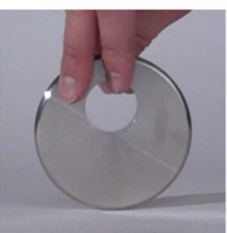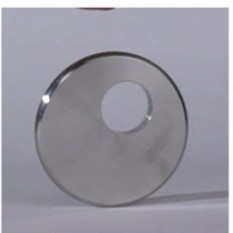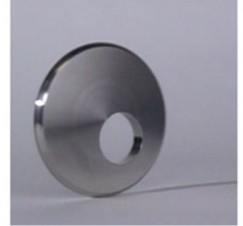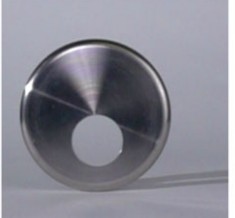

**Figure 2.** Tippedisk: inversion phenomenon. (Online version in colour.)

$\Omega > \Omega_{\mathrm{crit}}$ a stable inverted spinning solution. For subcritical spinning velocities $\Omega < \Omega_{\mathrm{crit}}$, the equilibrium associated with inverted spinning is unstable.

The overarching goal is to understand the qualitative dynamics behind the inversion behaviour of the tippedisk. Therefore, we aim to conduct an in-depth stability analysis based on the reduced model, derived in [19]. In this paper, a harmonic balance analysis is performed in order to characterize the Hopf bifurcation as sub- or supercritical. Moreover, the closed-form expressions are validated by a numerical shooting method. The structure of the system equations suggests the application of the theory of singular perturbations, indicating slow-fast system behaviour.

Section 2 briefly introduces the kinematics of the model derived in [19]. Furthermore, we provide the dimensions of the considered specimen and the reduced equations of motion. In §3, the local stability analysis of [19] is briefly repeated, raising the question of the type of Hopf bifurcation, which is answered subsequently. The nonlinear dynamical behaviour is studied in §4, visualized in §5 and discussed in §6.

## 2. Model of the tippedisk

In [18], a variety of different models, using various parametrizations and force laws, have been presented. With the aim to focus on the main physical effects, a reduced minimal model has been derived in [19], which forms the basis of the current paper. Before diving into the nonlinear dynamic analysis, we briefly review the kinematics of the reduced model from [19] to facilitate the transition to the present paper.

An orthonormal inertial frame $I = (O, \mathbf{e}_x^I, \mathbf{e}_y^I, \mathbf{e}_z^I)$ is introduced, attached to the origin $O$, such that $\mathbf{e}_z^I$ is perpendicular to a flat support. The body fixed $B$-frame $B = (G, \mathbf{e}_x^B, \mathbf{e}_y^B, \mathbf{e}_z^B)$ is located at the geometric centre $G$, so that $\mathbf{e}_z^B$ is normal to the surface of the disc. The unit vector $\mathbf{e}_x^B$ is in the direction of $\mathbf{r}_{GS}$, i.e. points from the geometric centre $G$ to the centre of gravity $S$. The disc is assumed to be in permanent contact with the support at the contact point $C$. For a more detailed description, we refer the reader to [18,19].

### (a) Dimensions and parameters

To be consistent with previous works [18,19], the dimensions and mass properties of the specimen under consideration are given in table 1. Here, the inertia tensor with respect to $G$, expressed in the body-fixed $B$-frame, is given as $_B\boldsymbol{\Theta}_G = \mathrm{diag}(A, B, C)$, where $B < A < C$ holds. To obtain more compact expressions, the variable $\bar{B}$ is introduced as $\bar{B} := B - m e^2$, which is equal to the moment of inertia $_B\boldsymbol{\Theta}_S(2,2)$ with respect to the centre of gravity $S$. The mass properties have been derived in detail in [18].

### (b) Equations of motion

In figure 3, the angles $\alpha$, $\beta$ and $\gamma$ define the orientation of the unbalanced disc, corresponding to Euler angles in the custom $z - x - z$ convention. It is worth mentioning here that the angles

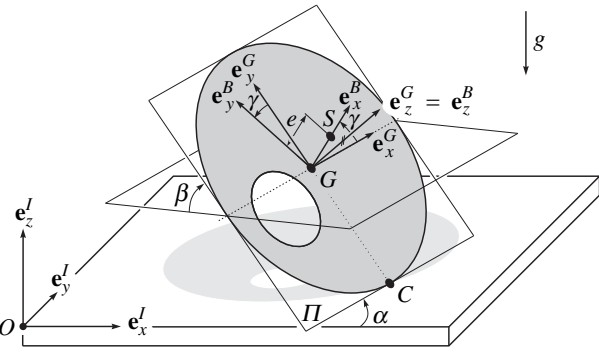

**Figure 3.** Mechanical model: tippedisk.

**Table 1.** Dimensions and mass properties of the tippedisk

| property | parameter | magnitude | unit |
|---|---|---|---|
| disc radius | $r$ | 0.045 | m |
| eccentricity | $e$ | $2.5 \times 10^{-3}$ | m |
| mass | $m$ | 0.435 | kg |
| $_B\boldsymbol{\Theta}_G(1,1)$ | $A$ | $0.249 \times 10^{-3}$ | kg m$^2$ |
| $_B\boldsymbol{\Theta}_G(2,2)$ | $B$ | $0.227 \times 10^{-3}$ | kg m$^2$ |
| $_B\boldsymbol{\Theta}_G(3,3)$ | $C$ | $0.468 \times 10^{-3}$ | kg m$^2$ |

correspond to an intrinsic parametrization, not to be confused with an extrinsic description. The angle $\alpha$ characterizes the rotation around the $\mathbf{e}_z^I$-axis. The angle $\beta$ describes the inclination of the disc, whereas $\gamma$ defines the relative angle between the grinding $G$-frame and body-fixed $B$-frame. In [19], it is shown that the spinning velocity $\dot{\alpha} = \Omega$ can be approximately assumed to be constant during the inversion of the disc, leading to a time evolution of the angle $\alpha$ expressed by the affine function

$$\alpha(t) = \Omega t + \alpha_0. \tag{2.1}$$

Without loss of generality, $\alpha_0$ can be set to zero. Introducing minimal coordinates $\mathbf{z} = [\beta, \gamma]^T$ and the scalar minimal velocity $\mathbf{v} = \dot{\beta}$, the dynamical behaviour of the tippedisk is described by the system of equations (see [19])

$$\dot{\mathbf{z}} = \mathbf{B}(\mathbf{z})\mathbf{v} + \boldsymbol{\beta}(\mathbf{z})$$
$$\mathbf{M}(\mathbf{z})\dot{\mathbf{v}} - \mathbf{h}(\mathbf{z}, \mathbf{v}) = \mathbf{f}_G + \mathbf{w}_y \lambda_{Ty}. \tag{2.2}$$

This reduced system in minimal coordinates $\mathbf{z} \in \mathbb{R}^2$ and minimal velocities $\mathbf{v} \in \mathbb{R}$ corresponds to a first-order ODE of total dimension three. The scalar mass matrix $\mathbf{M}$ and the vector of gyroscopic forces $\mathbf{h}$ are given as[1]

$$\mathbf{M} = A\cos^2\gamma + \bar{B}\sin^2\gamma + m(r + e\sin\gamma)^2\cos^2\beta \tag{2.3}$$

---

[1]For consistency with previous works, and to highlight the general structure, the quantities $\mathbf{M}$ and $\mathbf{h}$ as well as other quantities are written in bold, although they are scalar in this special case.

and

$$\mathbf{h} = +(A \cos^2 \gamma + \bar{B} \sin^2 \gamma)\Omega^2 \sin \beta \cos \beta - 2(A - \bar{B})\Omega \dot{\beta} \cos \beta \sin \gamma \cos \gamma$$
$$+ m(r + e \sin \gamma)^2 \dot{\beta}^2 \sin \beta \cos \beta + me(r + e \sin \gamma)\Omega^2 \sin \beta \cos^3 \beta \sin \gamma$$
$$- me(r + e \sin \gamma)(3 \sin^2 \beta - 2)\Omega \dot{\beta} \cos \beta \cos \gamma. \tag{2.4}$$

The generalized gravitational force

$$\mathbf{f}_G = -mg(r + e \sin \gamma) \cos \beta, \tag{2.5}$$

and generalized friction force $\mathbf{w}_y \lambda_{Ty}$ with corresponding force direction

$$\mathbf{w}_y = (r + e \sin \gamma) \sin \beta, \tag{2.6}$$

lateral sliding velocity

$$\gamma_y = (r + e \sin \gamma)\dot{\beta} \sin \beta - e\Omega \sin^2 \beta \cos \gamma, \tag{2.7}$$

and friction force $\lambda_{Ty}$ given by regularized Coulomb friction, also known as smooth Coulomb friction law

$$\lambda_{Ty} = -\mu mg \frac{\gamma_y}{|\gamma_y| + \varepsilon}, \tag{2.8}$$

form the right-hand side of equation (2.2). In the following analysis, we assume the linearized version of the smooth Coulomb friction law

$$\lambda_{Ty} = -\frac{\mu mg}{\varepsilon} \gamma_y, \tag{2.9}$$

to obtain more compact expressions. This assumption does not affect the qualitative dynamical behaviour. Assuming a linear friction law may seem artificial at this point, but its validity will be shown later in this paper. The friction coefficient is chosen as $\mu = 0.3$, the smoothing parameter is assumed to be $\varepsilon = 0.1 \, \text{m} \, \text{s}^{-1}$. The kinematic equations $(\beta)^{\cdot} = \dot{\beta}$ and $\dot{\gamma} = -\Omega \cos \beta$ are gathered in the first equation of system (2.2)

$$\dot{\mathbf{z}} = \mathbf{B}(\mathbf{z})\mathbf{v} + \boldsymbol{\beta}(\mathbf{z}, t), \tag{2.10}$$

with

$$\mathbf{B}(\mathbf{z}) = \begin{bmatrix} 1 \\ 0 \end{bmatrix} \quad \text{and} \quad \boldsymbol{\beta}(\mathbf{z}) = \begin{bmatrix} 0 \\ -\Omega \cos \beta \end{bmatrix}. \tag{2.11}$$

# 3. Local dynamics of the three-dimensional system

In [19], a linear stability analysis has been conducted in closed form that characterizes the stability of the inverted spinning solution, being an equilibrium of system (2.2). As we aim to analyse the qualitative behaviour in this paper, a brief summary of the results obtained in [19] is provided in §3a.

## (a) Linear stability analysis

As the tippedisk is called *inverted* when $\beta = +\pi/2$ and $\gamma = +\pi/2$ holds, new shifted coordinates

$$\bar{\mathbf{z}} := \begin{bmatrix} \bar{\beta} \\ \bar{\gamma} \end{bmatrix} := \begin{bmatrix} \beta - \dfrac{\pi}{2} \\ \gamma - \dfrac{\pi}{2} \end{bmatrix} \tag{3.1}$$

are introduced, such that the system equations (2.2), can be locally approximated by neglecting higher order terms of $\bar{\beta}$ and $\bar{\gamma}$. The linearization of the system (2.2) around the 'inverted spinning'

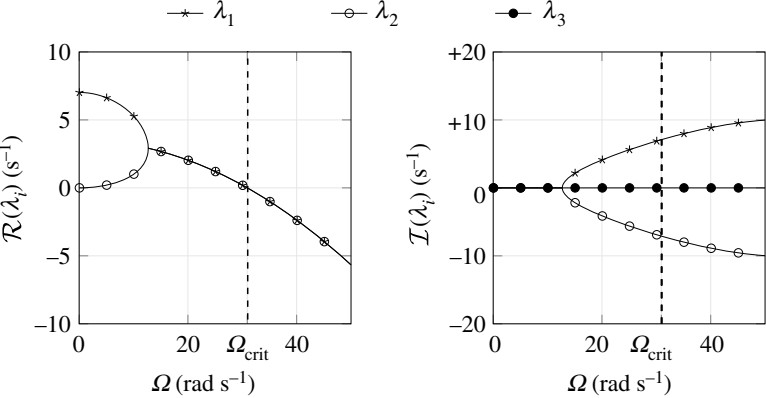

**Figure 4.** Eigenvalues for the inverted tippedisk for varying spinning velocity $\Omega$ [19]. The critical spinning velocity $\Omega_{\text{crit}}$ characterizes a Hopf bifurcation as a pair of two complex conjugate eigenvalues enters the right half of the complex plane.

equilibrium then yields the linear homogeneous system with constant coefficients

$$\dot{\mathbf{x}} = \begin{bmatrix} \dot{\bar{\beta}} \\ \dot{\bar{\gamma}} \\ \ddot{\bar{\beta}} \end{bmatrix} = \begin{bmatrix} 0 & 0 & 1 \\ \Omega & 0 & 0 \\ A_{31} & A_{32} & A_{33} \end{bmatrix} \begin{bmatrix} \bar{\beta} \\ \bar{\gamma} \\ \dot{\bar{\beta}} \end{bmatrix} = \mathbf{A}\mathbf{x}, \tag{3.2}$$

with

$$A_{31} = \frac{mg}{\bar{B}}(r+e) - \Omega^2 = \mathcal{O}(1)$$

$$A_{32} = -\frac{\mu mg}{\varepsilon \bar{B}}e(r+e)\Omega = \mathcal{O}(\tfrac{1}{\varepsilon}) \tag{3.3}$$

$$A_{33} = -\frac{\mu mg}{\varepsilon \bar{B}}(r+e)^2 = \mathcal{O}(\tfrac{1}{\varepsilon}).$$

As we see from equation (3.3), the matrix coefficient $A_{31}$ does not depend on the smoothing parameter $\varepsilon$ and is therefore of order $\mathcal{O}(1)$. Both $A_{32}$ and $A_{33}$ depend proportionally on $1/\varepsilon$ and are therefore of order $\mathcal{O}(1/\varepsilon)$. The non-inverted spinning is always unstable, whereas the stability of inverted spinning is characterized by the eigenvalues $\lambda_i$ for $i \in \{1, 2, 3\}$ of equation (3.2). The evolution of $\lambda_i$ is shown in figure 4 as a function of the spinning velocity $\Omega$. The real part of $\lambda_3$ is approximately given as

$$\lambda_3 = A_{33} + \mathcal{O}(\varepsilon) = -\frac{\mu mg}{\varepsilon \bar{B}}(r+e)^2 + \mathcal{O}(\varepsilon) \approx -129.04 \, \frac{1}{\text{s}}, \tag{3.4}$$

and therefore not shown in figure 4. For $\Omega = \Omega_{\text{crit}}$, a pair of complex conjugate eigenvalues is crossing the imaginary axis, indicating a Hopf bifurcation. If the spinning speed $\Omega$ is lower than the critical spinning velocity

$$\Omega_{\text{crit}} = \sqrt{\frac{(r+e)^2}{r}\frac{mg}{\bar{B}}} = 30.92 \, \text{rad s}^{-1}, \tag{3.5}$$

the inverted spinning solution is unstable. For supercritical spinning velocities inverted spinning becomes stable on 'fast' and 'intermediate' time scales. Perhaps somewhat unexpectedly, it turns out that the critical spinning velocity $\Omega_{\text{crit}}$, and thus the occurrence of the Hopf bifurcation, does not depend on the friction parameters $\mu$ and $\varepsilon$, see [19].

## (b) Harmonic balance method

To characterize the Hopf bifurcation as sub- or supercritical, we apply a harmonic balance method (HBM) to obtain a closed-form expression for the existence of the periodic solution. Isolating quartic orders $\mathcal{O}(||\bar{\mathbf{z}}||^4)$ in equation (2.2) by making use of

$$\sin\beta = \cos\bar{\beta} = 1 - \tfrac{1}{2}\bar{\beta}^2 + \mathcal{O}(\bar{\beta}^4), \tag{3.6}$$

$$\cos\beta = -\sin\bar{\beta} = -\bar{\beta} + \tfrac{1}{6}\bar{\beta}^3 + \mathcal{O}(\bar{\beta}^5), \tag{3.7}$$

$$\sin\gamma = \cos\bar{\gamma} = 1 - \tfrac{1}{2}\bar{\gamma}^2 + \mathcal{O}(\bar{\gamma}^4), \tag{3.8}$$

$$\cos\gamma = -\sin\bar{\gamma} = -\bar{\gamma} + \tfrac{1}{6}\bar{\gamma}^3 + \mathcal{O}(\bar{\gamma}^5), \tag{3.9}$$

yields the local approximation

$$\tilde{\mathbf{M}}(\bar{\mathbf{z}})\,\ddot{\bar{\beta}} - \tilde{\mathbf{h}}(\bar{\mathbf{z}}, \dot{\bar{\mathbf{z}}}) = \tilde{\mathbf{f}}(\bar{\mathbf{z}}, \dot{\bar{\mathbf{z}}}) + \mathcal{O}(||\bar{\mathbf{z}}||^4), \tag{3.10}$$

which corresponds to a scalar second order equation for the shifted inclination angle $\bar{\beta}$. To classify the nature of the Hopf bifurcation, the equations of motion equation (3.10) must be approximated at least up to cubic orders. The mass matrix $\tilde{\mathbf{M}}(\bar{\mathbf{z}})$ and vector of gyroscopic forces $\tilde{\mathbf{h}}(\bar{\mathbf{z}}, \dot{\bar{\mathbf{z}}})$ are given as

$$\tilde{\mathbf{M}}(\bar{\mathbf{z}}) = (A - \bar{B})\bar{\gamma}^2 + \bar{B} + m(r + e)^2\bar{\beta}^2 + \mathcal{O}(||\bar{\mathbf{z}}||^4) \tag{3.11}$$

and

$$\tilde{\mathbf{h}}(\bar{\mathbf{z}}, \dot{\bar{\mathbf{z}}}) = -\bar{B}\Omega^2\bar{\beta} + (\bar{B} - A)\Omega^2\bar{\beta}\bar{\gamma}^2 + \left[\frac{2}{3}\bar{B} - me(r + e)\right]\Omega^2\bar{\beta}^3$$
$$- 2[A - \bar{B} + me(r + e)]\Omega\dot{\bar{\beta}}\bar{\beta}\bar{\gamma} - m(r + e)^2\dot{\bar{\beta}}^2\bar{\beta} + \mathcal{O}(||\bar{\mathbf{z}}||^4). \tag{3.12}$$

The right-hand side of equation (3.10) is defined as generalized force $\tilde{\mathbf{f}} := \tilde{\mathbf{f}}_G + \tilde{\mathbf{w}}_y\lambda_{Ty}$, with

$$\tilde{\mathbf{f}}_G = +mg(r + e)\bar{\beta} - \frac{1}{2}mge\bar{\beta}\bar{\gamma}^3 - \frac{1}{6}mg(r + e)\bar{\beta}^3 + \mathcal{O}(||\bar{\mathbf{z}}||^4) \tag{3.13}$$

and

$$\tilde{\mathbf{w}}_y\lambda_{Ty} = -\frac{\mu mg}{\varepsilon}\left[-\Omega e\frac{4e + r}{6}\bar{\gamma}^3 + (r + e)^2\dot{\bar{\beta}} - \frac{3}{2}\Omega e(r + e)\bar{\beta}^2\bar{\gamma}\right.$$
$$\left. -(r + e)^2\dot{\bar{\beta}}\bar{\beta}^2 + e(r + e)\Omega\bar{\gamma} - e(r + e)\dot{\bar{\beta}}\bar{\gamma}^2\right] + \mathcal{O}(||\bar{\mathbf{z}}||^4). \tag{3.14}$$

According to the second row of equation (2.10), the kinematic relation is approximately given in terms of $\dot{\bar{\gamma}}$ and $\bar{\beta}$ as

$$\dot{\bar{\gamma}} = +\Omega\bar{\beta} + \mathcal{O}(|\bar{\beta}|^3). \tag{3.15}$$

If the harmonic ansatz

$$\bar{\beta} = C\sin(\omega t) \tag{3.16}$$

$$\bar{\gamma} = D\sin(\omega t + \varphi), \tag{3.17}$$

with amplitudes $C$, $D$, angular frequency $\omega$ and phase $\varphi$ is inserted into the kinematic relation equation (3.15), we obtain by coefficient comparison $\varphi = \frac{\pi}{2}$ and $D = -C\frac{\Omega}{\omega}$, yielding

$$\bar{\beta} = C\sin(\omega t) \quad \bar{\gamma} = -C\frac{\Omega}{\omega}\cos(\omega t)$$

$$\dot{\bar{\beta}} = C\omega\cos(\omega t) \quad \dot{\bar{\gamma}} = C\Omega\sin(\omega t) \tag{3.18}$$

$$\ddot{\bar{\beta}} = -C\omega^2\sin(\omega t),$$

where the identity $\cos(\omega t) = \sin(\omega t + \pi/2)$ is used and orders of $\mathcal{O}(|\bar{\beta}|^3)$ are neglected. Substitution of the harmonic ansatz in vectorial form

$$\hat{\mathbf{z}} = \begin{bmatrix} C\sin(\omega t) \\ -C\frac{\Omega}{\omega}\cos\omega t \end{bmatrix}, \tag{3.19}$$

into the quartic approximated system (3.10), leads to an equation of the form

$$-\hat{\mathbf{M}}(C,\omega)C\omega^2\sin(\omega t) = \hat{\mathbf{h}}(C,\omega) + \hat{\mathbf{f}}(C,\omega) + \mathcal{O}(C^4), \tag{3.20}$$

with mass matrix $\hat{\mathbf{M}}(C,\omega) := \tilde{\mathbf{M}}(\hat{\mathbf{z}})$, vector of gyroscopic forces $\hat{\mathbf{h}}(C,\omega) := \tilde{\mathbf{h}}(\hat{\mathbf{z}},\dot{\hat{\mathbf{z}}})$ and external forces $\hat{\mathbf{f}}(C,\omega) := \tilde{\mathbf{f}}(\hat{\mathbf{z}},\dot{\hat{\mathbf{z}}})$. Since equation (3.20) contains higher orders of trigonometric expressions $(\cos^2(\omega t), \sin^2(\omega t),\dots)$, we shift the exponents into the arguments by applying trigonometric addition theorems

$$\sin(\omega t)\cos^2(\omega t) = \frac{1}{4}\sin\omega t + \frac{1}{4}\sin 3\omega t \tag{3.21}$$

$$\sin^2(\omega t)\cos(\omega t) = \frac{1}{4}\cos\omega t - \frac{1}{4}\cos 3\omega t \tag{3.22}$$

$$\sin^3(\omega t) = \frac{3}{4}\sin\omega t - \frac{1}{4}\sin 3\omega t \tag{3.23}$$

$$\cos^3(\omega t) = \frac{3}{4}\cos\omega t + \frac{1}{4}\cos 3\omega t, \tag{3.24}$$

in harmonics of $\omega$. Neglecting higher harmonics in equation (3.20), the balance of $\sin(\omega t)$ and $\cos(\omega t)$ yields

$$\sin(\omega t): C^2\kappa_1 + \left[\bar{B}(\Omega^2 - \omega^2) - mg(r + e)\right] = 0 \tag{3.25}$$

and

$$\cos(\omega t): \frac{1}{4}C^2\frac{\kappa_2}{\omega} + (r + e)\left[e\frac{\Omega^2}{\omega} - (r + e)\omega\right] = 0, \tag{3.26}$$

with the parameters

$$\kappa_1 = \frac{1}{4}\left[\bar{B} - 3A - 2me(r + e)\right]\Omega^2 - \frac{1}{2}m(r + e)^2\omega^2 + \frac{1}{8}mge\frac{\Omega^2}{\omega^2} - \frac{1}{4}(\bar{B} - A)\frac{\Omega^4}{\omega^2} + \frac{1}{8}mg(r + e) \tag{3.27}$$

and

$$\kappa_2 = -e\frac{r + 4e}{2}\frac{\Omega^4}{\omega^2} + \frac{3}{2}e(r + e)\Omega^2 + (r + e)^2\omega^2. \tag{3.28}$$

If $\mathcal{O}(C^2)$ are neglected in equation (3.26), we obtain

$$\omega^2 = \frac{e}{r + e}\Omega^2 + \mathcal{O}(C^2), \tag{3.29}$$

which is constant with respect to orders $\mathcal{O}(C^2)$ and corresponds to the imaginary part of the critical eigenvalues $\lambda_{1,2} = \pm i\omega$. As the classification of the Hopf bifurcation depends on quadratic terms, $C^2$ cannot be neglected, so that the solution must be approximated up to higher order terms. Therefore, the correction term $\delta = \delta(C,\omega)$ is added, such that

$$\omega^2 = \frac{e}{r + e}\Omega^2 + \delta C^2 + \mathcal{O}(C^4), \tag{3.30}$$

describes the solution of equation (3.26) up to orders $\mathcal{O}(C^4)$. Multiplying equation (3.26) with the factor $4\omega$ and inserting equation (3.28) yields

$$C^2\left[-e\frac{r + 4e}{2(r + e)}\frac{\Omega^4}{\omega^2} + \frac{3}{2}e\Omega^2 + (r + e)\omega^2\right] + 4\left[e\Omega^2 - (r + e)\omega^2\right] = 0, \tag{3.31}$$

and the ansatz equation (3.30)

$$C^2\left[-\frac{r + 4e}{2}\Omega^2 + \frac{5}{2}e\Omega^2\right] - 4(r + e)\delta C^2 = \mathcal{O}(C^4), \tag{3.32}$$

from which $\delta$ is obtained up to second orders $\mathcal{O}(C^2)$ as

$$\delta = -\frac{1}{8}\frac{r-e}{r+e}\Omega^2 + \mathcal{O}(C^2). \tag{3.33}$$

Thus the solution

$$\omega^2 = \left(\frac{e}{r+e} - \frac{1}{8}\frac{r-e}{r+e}C^2\right)\Omega^2 + \mathcal{O}(C^4), \tag{3.34}$$

of equation (3.26) is given up to quartic orders $\mathcal{O}(C^4)$, which can be inserted into the balance of $\sin(\omega t)$ from equation (3.25)

$$C^2\kappa_1 + \left[\bar{B}\left(1 - \left(\frac{e}{r+e} - \frac{1}{8}\frac{r-e}{r+e}C^2\right)\right)\Omega^2 - mg(r+e)\right] = \mathcal{O}(C^4), \tag{3.35}$$

yielding the quadratic equation in amplitude $C$

$$C^2\left(\kappa_1(r+e) + \frac{1}{8}\bar{B}(r-e)\Omega^2\right) + [\bar{B}r\Omega^2 - mg(r+e)^2] = \mathcal{O}(C^4). \tag{3.36}$$

The amplitude $C$ follows in closed form as

$$C = 2\sqrt{\frac{e}{r+e}}\sqrt{-\frac{\bar{B}r\Omega^2 - mg(r+e)^2}{\chi\Omega^2 + mge(r+e)}} + \mathcal{O}(C^4) \tag{3.37}$$

with constant

$$\chi = A(r-2e) - \bar{B}\frac{2r^2 + e(r+e)}{2(r+e)} = 1.31 \times 10^{-7}\,\text{kg m}^3, \tag{3.38}$$

and exists, if the argument below the square root of equation (3.37) is greater than zero. As $\chi > 0$, the denominator $\chi\Omega^2 + mge(r+e)$ is positive for all spinning velocities $\Omega$, such that a real valued amplitude $C$ exists for

$$\Omega \leq \sqrt{\frac{mg}{\bar{B}}\frac{(r+e)^2}{r}} = \Omega_{\text{crit}}. \tag{3.39}$$

This condition of existence is in accordance with the critical spinning velocity $\Omega_{\text{crit}}$ derived in [19]. A branch of periodic solutions emerges at the bifurcation point, i.e. when the spinning speed $\Omega$ is equal to the critical spinning velocity $\Omega_{\text{crit}}$. Combining the knowledge of a Hopf bifurcation and the existence of periodic solutions for $\Omega \leq \Omega_{\text{crit}}$, the bifurcation at $\Omega_{\text{crit}}$ is characterized as a supercritical Hopf[2] bifurcation where stable periodic orbits coexist around an unstable equilibrium. The angular frequency $\omega$ of the periodic solution is obtained by inserting equation (3.37) into equation (3.34), which, neglecting $\mathcal{O}(C^4)$, yields

$$\omega = \sqrt{\frac{e}{r+e}\left(1 + \frac{1}{2}\frac{(r-e)}{(r+e)}\frac{\bar{B}r\Omega^2 - mg(r+e)^2}{\chi\Omega^2 + mge(r+e)}\right)}\Omega, \tag{3.40}$$

with corresponding period time $T = 2\pi/\omega$ of the periodic solution. The period time $T_{\text{crit}} = 0.886\,\text{s}$ at the bifurcation point, is calculated from equation (3.40) by inserting the critical spinning velocity $\Omega = \Omega_{\text{crit}}$. In figure 5a, the $\bar{\beta}$-amplitude $\bar{\beta}_{\text{max}} = C$ is depicted as a function of angular velocity $\Omega$. Furthermore, the equilibrium corresponding to the inverted steady-state solution is shown as a horizontal line $\bar{\beta}_{\text{max}} = 0$. Figure 5b shows the dependence of the period time $T$ under influence of the spinning velocity $\Omega$. Here, it is worth mentioning that for spinning velocities $\Omega > \Omega_{\text{crit}}$ there is no periodic solution and hence no period time $T$. The bifurcation point corresponding to the Hopf bifurcation is marked as a black dot. To illustrate the local validity of the closed-form solutions obtained with the HBM approach, the solutions are continued by dotted lines in each case.

---

[2]The name *supercritical Hopf* is somewhat misleading, since the branch of stable periodic solutions exists for 'subcritical' spinning velocities $\Omega \leq \Omega_{\text{crit}}$. Nevertheless, we stick to this terminology as it is common in classic literature [20,21]

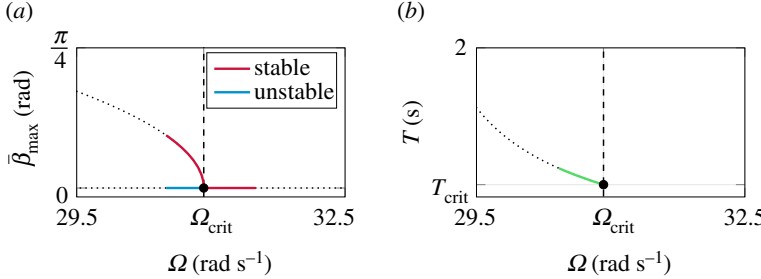

**Figure 5.** Harmonic balance method (HBM) results around $\Omega_{crit}$. (*a*) Bifurcation diagram obtained from HBM in closed form, depicting the supercritical Hopf bifurcation and the birth of a branch with stable limit cycles. (*b*) Closed-form approximation of the period time of the limit cycle. (Online version in colour.)

## 4. Nonlinear dynamics

In the previous section, the dynamics is studied by closed-form expressions coming from a linearization and the HBM from equation (3.18). The search for closed-form expressions necessitates local approximations of the dynamics by neglecting higher order terms in $C$ as well as higher harmonics. For this reason, the single harmonic balance result is valid only for small amplitude $C$ and thus near the bifurcation point. Of course, a more accurate result could have been obtained with a multi-HBM without approximations, but this would not be possible in closed form. As we are not only interested in the local dynamics near $\Omega \approx \Omega_{crit}$, in this section we identify periodic solutions using the numerical shooting method in combination with continuation in $\Omega$. Moreover, we exploit the singularly perturbed structure of the system equations to gain a deeper understanding of the qualitative behaviour behind the inversion of the tippedisk.

### (a) Continuation of periodic solutions

The shooting method [21,22] combined with a continuation method [23] is a popular approach to construct a numerical bifurcation diagram. However, a direct application of these classical numerical methods to the problem of the tippedisk leads to convergence problems as the singularly perturbed structure of the system equations results in an extremely stiff set of ODEs. In particular, more elaborate variants of these methods, such as the multiple shooting method and arclength continuation with variable stepsize, proved to be prone to convergence problems. For completeness, we briefly review the adopted shooting method and sequential continuation technique together with the chosen modifications to guarantee a robust continuation. The basic idea behind the classical sequential shooting method is to change the bifurcation parameter sequentially and formulate a zero-finding problem that can be solved by a Newton-type algorithm. Here, the spinning velocity $\Omega$ is chosen as a bifurcation parameter. The single shooting method formulates a two-point boundary value problem in terms of a zero-finding problem that can be solved with Newton-like methods. For an autonomous nonlinear system of the form

$$\dot{\mathbf{q}} = \mathbf{F}(\mathbf{q}) \in \mathbb{R}^3, \tag{4.1}$$

the two-point boundary value problem consists of the periodicity condition

$$\mathbf{r}_p(\mathbf{q}_0, T) = \boldsymbol{\varphi}(\mathbf{q}_0, t_0 + T) - \mathbf{q}_0 = \int_{t_0}^{t_0+T} \mathbf{F}(\mathbf{q}(\tau))\, \mathrm{d}\tau \in \mathbb{R}^3, \tag{4.2}$$

together with a suitable anchor equation as the period time is *a priori* unknown. For the reduced model of the tippedisk, we have the state vector $\mathbf{q} = [\mathbf{z}, \mathbf{v}]^{\mathrm{T}} \in \mathbb{R}^3$ and the most robust results can

**Table 2.** Initial guess for sequential continuation

| estimated quantity | magnitude | unit |
|:---:|:---:|:---:|
| $\bar{\beta}_0$ | 0 | rad |
| $\bar{\gamma}_0$ | 1.69 | rad |
| $\dot{\bar{\beta}}_0$ | −1.72 | rad s$^{-1}$ |
| $T_0$ | 1.10 | s |

be obtained by choosing the simple anchor

$$\mathbf{r}_a(\mathbf{q}_0, T) = \bar{\beta}_0 \in \mathbb{R}. \tag{4.3}$$

The combination of the periodicity residual $\mathbf{r}_p$ and the anchor equation $\mathbf{r}_a$ yields the four-dimensional residuum

$$\mathbf{r}(\mathbf{q}_0, T) := \begin{bmatrix} \mathbf{r}_p(\mathbf{q}_0, T) \\ \mathbf{r}_a(\mathbf{q}_0, T) \end{bmatrix} \in \mathbb{R}^4. \tag{4.4}$$

Periodic solutions are associated with the zeros of the residuum $\mathbf{r}(\mathbf{q}_0, T) = 0$, which specifies a state $\mathbf{q}_0$ on the $T$-periodic solution. To solve the zero-finding problem, any standard Newton-type algorithm can be applied, starting with an initial guess $(\mathbf{q}_0^{(0)}, T_0^{(0)})$ and resulting in the converged solution $(\mathbf{q}_0^{(*)}, T_0^{(*)})$. The dependence on the spinning velocity $\Omega$ is studied by a sequential continuation method, where $\Omega_i$ is an element of the set $\mathcal{A} = \{\Omega_0, \Omega_1, \ldots, \Omega_n\}$ and the index $i \in \mathbb{N}$ is incremented stepwise. Sequential continuation combines a predictor step, where the initial estimate $(\mathbf{q}_0^{(0),i}, T_0^{(0),i})$ for a given $\Omega_i$ comes from the solution $(\mathbf{q}_0^{(*),i-1}, T_0^{(*),i-1})$ of the shooting problem at $\Omega_{i-1}$ with a subsequent corrector step, viz. the shooting procedure. The initial estimate of the periodic solution for $\Omega_0 = \Omega_{\mathrm{crit}} - 0.5 \, \mathrm{rad \, s}^1$ is given in table 2 and defines the starting point for the sequential continuation. To track the evolution of periodic orbits, in a first step the spinning speed $\Omega$ is increased to analyse the behaviour near the Hopf bifurcation at $\Omega_{\mathrm{crit}}$. The corresponding increasing set $\mathcal{A}^{\mathrm{in}}$ is chosen as

$$\mathcal{A}^{\mathrm{in}} = \{\Omega_{i+1} \in \mathbb{R} | \Omega_{i+1} = \Omega_i + (\Omega_{\mathrm{crit}} - \Omega_i)/100, \ i \in \mathcal{I}\}, \tag{4.5}$$

with respect to the index set $\mathcal{I} = \{1, 2, \ldots, 400\}$. In a second step, the behaviour for decreasing $\Omega$ is analysed by defining the decreasing $\Omega$-set

$$\mathcal{A}^{\mathrm{de}} = \{\Omega_{i+1} \in \mathbb{R} | \Omega_{i+1} = \Omega_i + (\Omega_{\mathrm{h}} - \Omega_i)/100, \ i \in \mathcal{I}\}. \tag{4.6}$$

Note both sets $\mathcal{A}^{\mathrm{in}}$ and $\mathcal{A}^{\mathrm{de}}$ are generated by convergent sequences, which results in a fine resolution around $\Omega_{\mathrm{crit}}$ and $\Omega_{\mathrm{h}}$. The spinning speed $\Omega_{\mathrm{crit}}$ corresponds to the critical spinning velocity at the Hopf bifurcation.

In figure 6, the branch of limit cycles obtained numerically with the adapted shooting-continuation method is shown. For comparison, the closed-form solutions obtained by the harmonic balance approach are depicted in black. According to the sequential shooting results, the bifurcation is identified as supercritical Hopf, since a stable periodic solution exists for $\Omega < \Omega_{\mathrm{crit}}$. For decreasing spinning velocities, the periodic solution vanishes at $\Omega_{\mathrm{h}} = 30.07 \, \mathrm{rad \, s}^{-1}$, with corresponding period time $T_h = \infty$. At this point, $\Omega_{\mathrm{h}}$ is not yet defined, but will be identified as the heteroclinic/homoclinic spinning speed in the following.

## (b) Singularly perturbed dynamics

In [19], it is shown that the dynamics of system (2.2) must be considered on different time scales. Before analysing the dynamical behaviour on the tippedisk in the framework of slow–fast systems, we introduce the basics of singular perturbation theory [24–26].

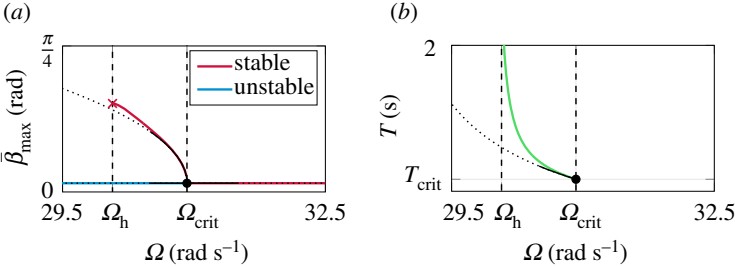

**Figure 6.** Numerical results from the shooting-continuation method. For comparison, the closed-form HBM approximations are shown in black. (*a*) bifurcation diagram, (*b*) period time. (Online version in colour.)

### (i) Basics of singular perturbation theory

Singular perturbation theory deals in the context of dynamics with systems of the form

$$\dot{\mathbf{x}} = \mathbf{f}(\mathbf{x}, \mathbf{y}; \varepsilon)$$
$$\varepsilon \dot{\mathbf{y}} = \mathbf{g}(\mathbf{x}, \mathbf{y}; \varepsilon),$$

(4.7)

where $\varepsilon \ll 1$ is identified as small fixed perturbation parameter and $\bullet := \frac{\mathrm{d}}{\mathrm{d}t}\bullet$ denotes the derivative with respect to 'slow' time $t$. The system

$$\dot{\mathbf{x}} = \mathbf{f}(\mathbf{x}, \mathbf{y}; \varepsilon) \in \mathbb{R}^n,$$

(4.8)

is called the slow subsystem with the corresponding slow variable $\mathbf{x} \in \mathbb{R}^n$, while the fast subsystem is identified as

$$\varepsilon \dot{\mathbf{y}} = \mathbf{g}(\mathbf{x}, \mathbf{y}; \varepsilon) \in \mathbb{R}^m,$$

(4.9)

with associated fast variable $\mathbf{y} \in \mathbb{R}^m$. By introducing the 'fast' time variable $\tau := \frac{1}{\varepsilon}t$ and the associated derivative $\bullet' := \frac{\mathrm{d}}{\mathrm{d}\tau}\bullet$, the rescaled dynamical system is given by the differential equation

$$\mathbf{x}' = \varepsilon \, \mathbf{f}(\mathbf{x}, \mathbf{y}; \varepsilon)$$
$$\mathbf{y}' = \mathbf{g}(\mathbf{x}, \mathbf{y}; \varepsilon).$$

(4.10)

Setting the perturbation parameter $\varepsilon$ in equation (4.7) to zero, yields the critical system

$$\dot{\mathbf{x}} = \mathbf{f}(\mathbf{x}, \mathbf{y}; 0)$$
$$0 = \mathbf{g}(\mathbf{x}, \mathbf{y}; 0),$$

(4.11)

which corresponds to a differential algebraic equation on the slow time-scale $t$. According to the implicit function theorem, the algebraic equation $\mathbf{g}(\mathbf{x}, \mathbf{y}; 0) = 0$ can locally (i.e. in a neighbourhood $\mathcal{U}$ of $\bar{\mathbf{x}}$ with $\mathbf{g}(\mathbf{x}, \mathbf{y}; \varepsilon) = 0$) be cast in explicit form $\mathbf{y} = \mathbf{h}_c(\mathbf{x})$ if the Jacobian $\frac{\partial \mathbf{g}}{\partial \mathbf{y}}|_{\bar{\mathbf{x}}, \bar{\mathbf{y}}; 0}$ is invertible. This relation $\mathbf{y} = \mathbf{h}_c(\mathbf{x})$ between $\mathbf{y}$ and $\mathbf{x}$ describes the behaviour of the fast coordinate $\mathbf{y}$ induced by the evolution of the slow variable $\mathbf{x}$, defining the $n$-dimensional critical manifold

$$\mathcal{M}_c := \{(\mathbf{x}, \mathbf{y}) \in \mathbb{R}^{n+m} \mid \mathbf{y} = \mathbf{h}_c(\mathbf{x}), \ \mathbf{x} \in \mathcal{U}\}.$$

(4.12)

The dynamical behaviour on this critical manifold is characterized by the differential equation

$$\dot{\mathbf{x}} = \mathbf{f}(\mathbf{x}, \mathbf{h}_c(\mathbf{x}); 0).$$

(4.13)

Equivalently, equation (4.10) with $\varepsilon = 0$ gives the critical boundary layer system

$$\mathbf{x}' = \mathbf{0}$$
$$\mathbf{y}' = \mathbf{g}(\mathbf{x}, \mathbf{y}; 0),$$

(4.14)

which, with respect to the fast time-scale $\tau$, implies on the one hand a constant slow variable $\mathbf{x} = \mathbf{x}^*$ and on the other hand that $\mathbf{y}^* = \mathbf{h}_c(\mathbf{x}^*)$ is an equilibrium point, i.e. an element of the critical manifold $\mathcal{M}_c$. For $\varepsilon \neq 0$, the fast system indicates the equilibrium condition $\mathbf{g}(\mathbf{x}, \mathbf{y}; \varepsilon) = 0$, which implies the relation $\mathbf{y} = \mathbf{h}_s(\mathbf{x}; \varepsilon)$, which defines the corresponding $n$-dimensional slow invariant manifold

$$\mathcal{M}_s := \{(\mathbf{x}, \mathbf{y}) \in \mathbb{R}^{n+m} \mid \mathbf{y} = \mathbf{h}_s(\mathbf{x}; \varepsilon), \ \mathbf{x} \in \mathcal{U}\}. \tag{4.15}$$

The form of the slow manifold $\mathcal{M}_s$ can be obtained through a perturbation technique by exploiting its invariance. Inserting $\mathbf{y} = \mathbf{h}_s(\mathbf{x}; \varepsilon)$ and $\dot{\mathbf{y}} = (\partial \mathbf{h}_s / \partial \mathbf{x}|_{\mathbf{x};\varepsilon})\dot{\mathbf{x}}$ into the fast dynamics (4.9) yields

$$\varepsilon \frac{\partial \mathbf{h}_s}{\partial \mathbf{x}}\bigg|_{\mathbf{x};\varepsilon} \mathbf{f}(\mathbf{x}, \mathbf{h}_s(\mathbf{x}); \varepsilon) = \mathbf{g}(\mathbf{x}, \mathbf{h}_s(\mathbf{x}); \varepsilon), \tag{4.16}$$

which is expanded using the convergent series

$$\mathbf{h}_s(\mathbf{x}; \varepsilon) = \mathbf{h}_0(\mathbf{x}) + \mathbf{h}_1(\mathbf{x})\varepsilon + \mathcal{O}(\varepsilon^2), \tag{4.17}$$

up to orders of $\mathcal{O}(\varepsilon^2)$ into equation

$$\varepsilon \left[ \frac{\partial \mathbf{h}_0(\mathbf{x})}{\partial \mathbf{x}}\bigg|_{\mathbf{x}} + \frac{\partial \mathbf{h}_1(\mathbf{x})}{\partial \mathbf{x}}\bigg|_{\mathbf{x}} \varepsilon \right] \left[ \mathbf{f}(\mathbf{x}, \mathbf{h}_0; 0) + \left( \frac{\partial \mathbf{f}}{\partial \varepsilon}\bigg|_{\mathbf{x}, \mathbf{h}_0; 0} + \frac{\partial \mathbf{f}}{\partial \mathbf{y}}\bigg|_{\mathbf{x}, \mathbf{h}_0; 0} \mathbf{h}_1 \right) \varepsilon \right]$$

$$= \mathbf{g}(\mathbf{x}, \mathbf{h}_0(\mathbf{x}); 0) + \left[ \frac{\partial \mathbf{g}}{\partial \varepsilon}\bigg|_{\mathbf{x}, \mathbf{h}_0; 0} + \frac{\partial \mathbf{g}}{\partial \mathbf{y}}\bigg|_{\mathbf{x}, \mathbf{h}_0; 0} \mathbf{h}_1 \right] \varepsilon + \mathcal{O}(\varepsilon^2). \tag{4.18}$$

Comparing the coefficients of powers of $\varepsilon$ yields

$$\varepsilon^0: \quad 0 = \mathbf{g}(\mathbf{x}, \mathbf{h}_0; 0) \tag{4.19}$$

$$\varepsilon^1: \quad \frac{\partial \mathbf{h}_0}{\partial \mathbf{x}}\bigg|_{\mathbf{x}} \mathbf{f}(\mathbf{x}, \mathbf{h}_0; 0) = \frac{\partial \mathbf{g}}{\partial \varepsilon}\bigg|_{\mathbf{x}, \mathbf{h}_0; 0} + \frac{\partial \mathbf{g}}{\partial \mathbf{y}}\bigg|_{\mathbf{x}, \mathbf{h}_0; 0} \mathbf{h}_1(\mathbf{x}) \tag{4.20}$$

$$\vdots$$

From equation (4.19), we conclude that $\mathbf{h}_0(\mathbf{x}) = \mathbf{h}_c(\mathbf{x})$, which indicates that the critical manifold $\mathcal{M}_c$ is equal to the zero-order approximation of the slow manifold $\mathcal{M}_s$. If $\partial \mathbf{g}/\partial \mathbf{y}|_{\mathbf{x}, \mathbf{h}_0; 0}$ is invertible, $\mathbf{h}_1(\mathbf{x})$ is deduced from equation (4.20) as

$$\mathbf{h}_1(\mathbf{x}) = \frac{\partial \mathbf{g}}{\partial \mathbf{y}}\bigg|_{\mathbf{x}, \mathbf{h}_0; 0}^{-1} \left[ \frac{\partial \mathbf{h}_0}{\partial \mathbf{x}}\bigg|_{\mathbf{x}} \mathbf{f}(\mathbf{x}, \mathbf{h}_0; 0) - \frac{\partial \mathbf{g}}{\partial \varepsilon}\bigg|_{\mathbf{x}, \mathbf{h}_0; 0} \right]. \tag{4.21}$$

If this procedure is continued to compute $\mathbf{h}_2, \mathbf{h}_3, \ldots, \mathbf{h}_n$, the slow manifold $\mathcal{M}_s$ can be approximated up to arbitrary orders $\mathcal{O}(\varepsilon^{n+1})$. The distance function to the slow manifold

$$\mathbf{d} := \mathbf{y} - \mathbf{h}_s(\mathbf{x}; \varepsilon), \tag{4.22}$$

i.e. $\mathbf{y} \in \mathcal{M}_s \Leftrightarrow \mathbf{d} = 0$, is governed by the fast dynamics

$$\mathbf{d}' = \mathbf{y}' - \frac{\partial \mathbf{h}_s}{\partial \mathbf{x}}\mathbf{x}' = \mathbf{y}' + \mathcal{O}(\varepsilon) = \mathbf{g}(\mathbf{x}, \mathbf{d} + \mathbf{h}_s(\mathbf{x}; \varepsilon); \varepsilon) + \mathcal{O}(\varepsilon). \tag{4.23}$$

Linearizing the distance dynamics around the slow manifold $\mathcal{M}_s$, i.e. $\mathbf{d} = 0$, yields

$$\mathbf{d}' = \underbrace{\mathbf{g}(\mathbf{x}, \mathbf{h}_0(\mathbf{x}), 0)}_{=0} + \frac{\partial \mathbf{g}}{\partial \mathbf{y}}\bigg|_{\mathbf{x}, \mathbf{h}_0; 0} \mathbf{d} = \frac{\partial \mathbf{g}}{\partial \mathbf{y}}\bigg|_{\mathbf{x}, \mathbf{h}_0; 0} \mathbf{d}, \tag{4.24}$$

neglecting orders $\mathcal{O}(\varepsilon)$. According to Lyapunov's indirect method, the slow manifold is locally attractive, if $\partial \mathbf{g}/\partial \mathbf{y}|_{\mathbf{x},\mathbf{h}_0;0}$ is Hurwitz. If the slow manifold is attractive, solutions converge on the fast time-scale to the slow manifold $\mathcal{M}_s$. The slow and thus the asymptotic behaviour is then governed by the dynamics on the slow manifold $\mathcal{M}_s$, indicating the reduction to the $n$-dimensional system

$$\mathbf{h}_s(\mathbf{x};\varepsilon) \approx \mathbf{h}_0(\mathbf{x}) + \varepsilon \mathbf{h}_1(\mathbf{x})$$
$$\dot{\mathbf{x}} = \mathbf{f}(\mathbf{x},\mathbf{h}_s(\mathbf{x};\varepsilon);\varepsilon),$$

(4.25)

neglecting orders $\mathcal{O}(\varepsilon^2)$.

## (ii) Singularly perturbed dynamics of the tippedisk

In this section, the singular perturbation theory presented in §4b(i) is applied to the reduced model of the tippedisk. Introducing the slow variables $\mathbf{x} = [\beta, \gamma]^{\mathrm{T}}$ and the fast variable $\mathbf{y} = \eta = \dot{\beta}$, we obtain the singularly perturbed system

$$\dot{\mathbf{x}} = \mathbf{f}(\mathbf{x},\mathbf{y})$$
$$\varepsilon \dot{\mathbf{y}} = \mathbf{g}(\mathbf{x},\mathbf{y};\varepsilon) = \mathbf{g}_0(\mathbf{x},\mathbf{y}) + \mathbf{g}_1(\mathbf{x},\mathbf{y})\varepsilon,$$

(4.26)

with

$$\mathbf{f}(\mathbf{x},\mathbf{y}) = \begin{bmatrix} \eta \\ -\Omega\cos\beta \end{bmatrix} \in \mathbb{R}^2,$$

(4.27)

$$\mathbf{g}_0(\mathbf{x},\mathbf{y}) = -\mathbf{M}^{-1}\mu m g\, \mathbf{w}_y(\mathbf{x})\gamma_y(\mathbf{x},\mathbf{y}) \in \mathbb{R},$$

(4.28)

and
$$\mathbf{g}_1(\mathbf{x},\mathbf{y}) = \mathbf{M}^{-1}[\mathbf{h}(\mathbf{x},\mathbf{y}) + \mathbf{f}_G(\mathbf{x},\mathbf{y})] \in \mathbb{R},$$

(4.29)

by normalizing and pre-multiplying equation (2.2) with the 'small' smoothing coefficient $\varepsilon > 0$ of the friction law, cf. [25]. The fast subsystem is given as

$$\varepsilon \dot{\mathbf{y}} = \mathbf{g}(\mathbf{x},\mathbf{y};\varepsilon) = \mathbf{g}_0(\mathbf{x},\mathbf{y}) + \mathbf{g}_1(\mathbf{x},\mathbf{y})\varepsilon.$$

(4.30)

For $\varepsilon = 0$, the fast subsystem collapses to the algebraic equation $\mathbf{g}_0(\mathbf{x},\mathbf{y}) = 0$, which according to equation (4.28) states that the relative velocity $\gamma_y(\mathbf{x},\mathbf{y})$ vanishes, i.e. the contact point of the tippedisk is in a state of pure rolling. Since the relative velocity $\gamma_y(\mathbf{x},\mathbf{y})$ depends linearly on the fast variable $\eta = \dot{\beta}$, the critical manifold exists globally as the Jacobian $\partial \mathbf{g}_0/\partial \mathbf{y}|_{\mathbf{x},\mathbf{y}}$ is invertible. The associated critical manifold $\mathcal{M}_c$ is given as

$$\mathcal{M}_c := \left\{ (\mathbf{x},\mathbf{y}) \in \mathbb{R}^3 \,|\, \mathbf{y} = \mathbf{h}_c(\mathbf{x}) = \frac{e\sin\beta\cos\gamma}{(r+e\sin\gamma)}\Omega, \mathbf{x} \in \mathbb{R}^2 \right\},$$

(4.31)

being the zero-order approximation of the slow manifold, which is given up to orders $\mathcal{O}(\varepsilon^2)$ as

$$\mathcal{M}_s := \left\{ (\mathbf{x},\mathbf{y}) \in \mathbb{R}^3 \,|\, \mathbf{y} = \frac{e\sin\beta\cos\gamma}{(r+e\sin\gamma)}\Omega + \mathbf{h}_1(\mathbf{x})\varepsilon + \mathcal{O}(\varepsilon^2), \mathbf{x} \in \mathbb{R}^2 \right\},$$

(4.32)

with

$$\mathbf{h}_1(\mathbf{x}) = \left.\frac{\partial \mathbf{g}}{\partial \mathbf{y}}\right|_{\mathbf{x},\mathbf{h}_c}^{-1} \left[ \left.\frac{\partial \mathbf{h}_c}{\partial \mathbf{x}}\right|_{\mathbf{x}} \mathbf{f}(\mathbf{x},\mathbf{h}_c) - \mathbf{g}_1(\mathbf{x},\mathbf{h}_c) \right].$$

(4.33)

The stability of the slow manifold, characterized by the distance dynamics

$$\mathbf{d}' = \left.\frac{\partial \mathbf{g}_0}{\partial \mathbf{y}}\right|_{\mathbf{x},\mathbf{h}_c} \mathbf{d},$$

(4.34)

is asymptotically stable, since the Jacobian

$$\left.\frac{\partial \mathbf{g}_0}{\partial \mathbf{y}}\right|_{\mathbf{x}, \mathbf{h}_c} = -\mathbf{M}^{-1} \mu m g \, (r + e \sin \gamma)^2 \sin^2 \beta, \tag{4.35}$$

is strictly negative for all $\beta \in (0, +\pi)$, i.e. in a basin of attraction orbits are attracted to the invariant manifold $\mathcal{M}_s$. Therefore, the asymptotic behaviour of attracted solutions is governed by the reduced two-dimensional system

$$\mathbf{h}_s(\mathbf{x}) \approx \mathbf{h}_c(\mathbf{x}) + \varepsilon \mathbf{h}_1(\mathbf{x})$$
$$\dot{\mathbf{x}} = \mathbf{f}(\mathbf{x}, \mathbf{h}_s(\mathbf{x})), \tag{4.36}$$

neglecting orders $\mathcal{O}(\varepsilon^2)$.

## 5. Dynamics on the slow manifold

After having analysed the qualitative behaviour of the system in the previous sections, we will now depict the dynamical behaviour on the slow manifold embedded in the three-dimensional state space. According to the linear stability analysis of [19], equilibria corresponding to 'non-inverted spinning' are unstable. The 'inverted spinning' equilibrium is unstable for $\Omega < \Omega_{\text{crit}}$ and stable for supercritical spinning velocities $\Omega > \Omega_{\text{crit}}$. Due to the ambiguity of trigonometric expressions, we find that both the state $\mathbf{x}_{\text{non}} = (+\pi/2, -\pi/2, 0)$ and the state $\mathbf{x}_{\text{non}} = (+3\pi/2, +\pi/2, 0)$ correspond to an equilibrium that is associated as non-inverted spinning (alternatively we may employ a cylindrical state space). Inverted spinning is characterized by the equilibrium $\mathbf{x}_{\text{in}} = (+\pi/2, +\pi/2, 0)$.

Figure 7 shows the behaviour of trajectories in $\beta$-$\gamma$-$\dot{\beta}$ state space under variation of the spinning speed $\Omega$. The associated discrete spinning velocities are shown in figure 8, where the dots represent the inverted spinning equilibrium, and the square marks correspond to periodic solutions. The slow manifold $\mathcal{M}_s$, defined in equation (4.15), is depicted as grey surface in figure 7. Unstable equilibria are shown as blue dots, stable ones as red dots. For each subfigure, two orbits, initialized as black crosses at $\mathbf{x}_0^1 = (+\pi/2, -\pi/2, 2)$ and $\mathbf{x}_0^2 = (+\pi/2, +\pi/2 + 0.4, 0)$, are shown as cyan trajectories. For $\Omega < \Omega_{\text{h}}$, solutions are repelled by the inverted spinning equilibrium (figure 7a). At $\Omega_{\text{h}}$, a periodic solution (with period time $T = \infty$) arises that includes both non-inverted spinning equilibria. For $\Omega_{\text{h}} < \Omega < \Omega_{\text{crit}}$, this periodic solution attracts both orbits and shrinks for increasing $\Omega$ (figure 7b–e). At $\Omega_{\text{crit}}$, the periodic solution collapses, such that the inverted spinning equilibrium becomes stable (depicted as a red dot) and attracts the initialized trajectories (figure 7f). In addition, we observe that all trajectories converge rapidly onto the slow manifold $\mathcal{M}_s$. After convergence, the orbits evolve on this two-dimensional manifold. Due to this attractivity and the resulting reduced two-dimensional behaviour, it is possible to project the three-dimensional dynamics in figure 9 onto the $(\beta, \gamma)$-plane to obtain a clearer representation without losing too much information.

## 6. Discussion

In the vicinity of the bifurcation point at $\Omega_{\text{crit}}$, the results of sequential shooting and the harmonic balance approach agree, showing the validity of the HBM at the bifurcation point $\Omega = \Omega_{\text{crit}}$. Together with results from [19], the bifurcation at

$$\Omega_{\text{crit}} = \sqrt{\frac{(r+e)^2}{r} \frac{mg}{\bar{B}}} = 30.92 \, \text{rad s}^{-1}, \tag{6.1}$$

is characterized as supercritical Hopf bifurcation where a stable periodic solution collapses with an unstable equilibrium, resulting in a stable equilibrium for $\Omega > \Omega_{\text{crit}}$. For significantly subcritical spinning velocities $\Omega < \Omega_{\text{crit}}$, the amplitude and period time determined from numerical shooting and approximated closed-form harmonic balance differ increasingly with the

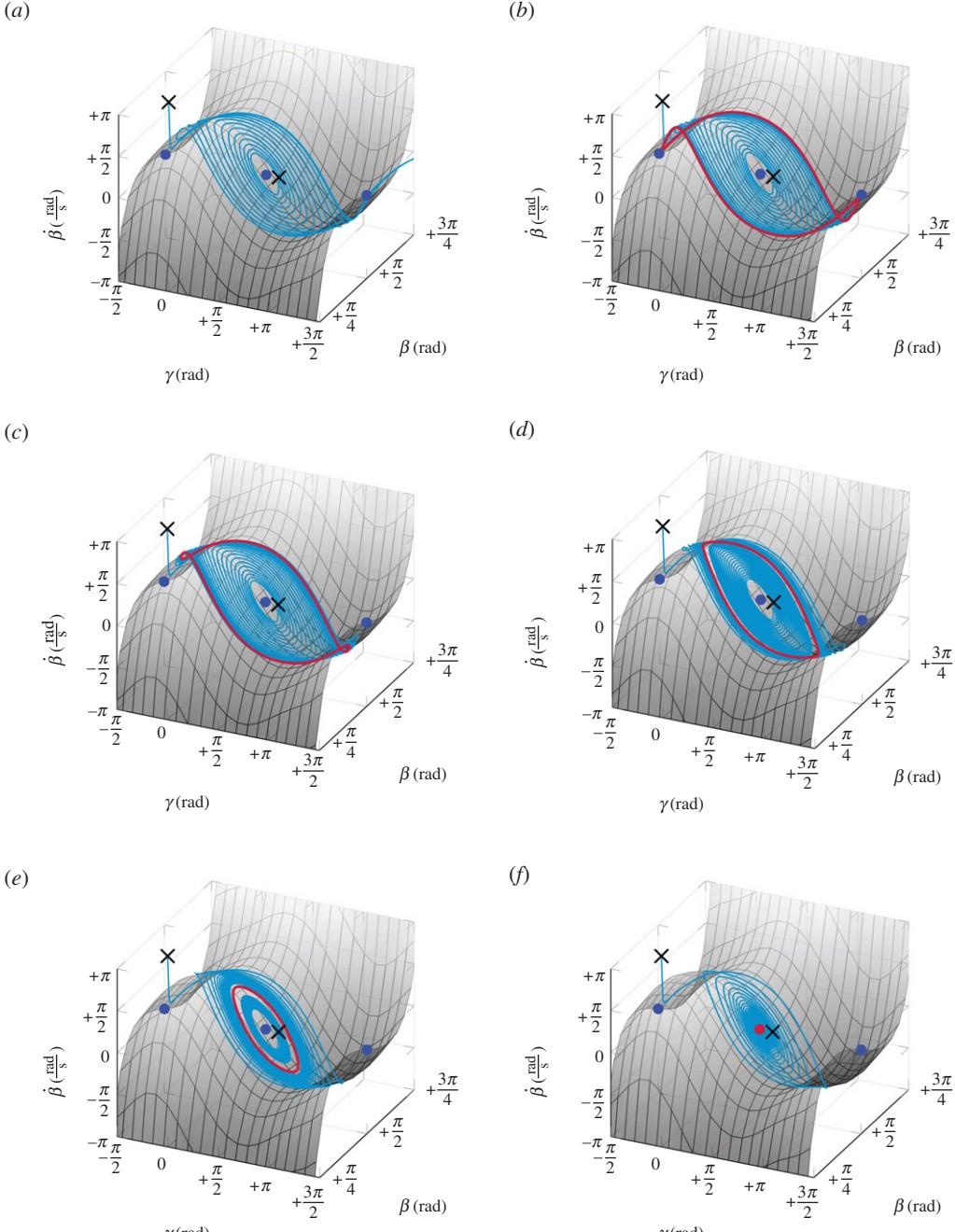

**Figure 7.** Dynamics of the tippedisk for the three-dimensional reduced model for various values of the spinning speed $\Omega$. The grey surface corresponds to the slow manifold $\mathcal{M}_s$. (a) $\Omega = \Omega_h - 0.1\,\mathrm{rad\ s^{-1}}$, (b) $\Omega = \Omega_h + 0.01\,\mathrm{rad\ s^{-1}} \approx \Omega_h$, (c) $\Omega = \Omega_h + 0.1\,\mathrm{rad\ s^{-1}}$, (d) $\Omega = \Omega_{crit} - 0.5\,\mathrm{rad\ s^{-1}}$, (e) $\Omega = \Omega_{crit} - 0.2\,\mathrm{rad\ s^{-1}}$, (f) $\Omega = \Omega_{crit} + 0.5\,\mathrm{rad\ s^{-1}}$. (Online version in colour.)

distance from the Hopf bifurcation. According to the results of numerical shooting, the periodic solution vanishes at $\Omega = \Omega_h$, with a corresponding infinite period time $T_h = \infty$. Section 4b(ii) discusses the singularly perturbed structure of the system, indicating an attractive slow manifold $\mathcal{M}_s$. To obtain more compact expressions, a linearized version of the regularized Coulomb friction

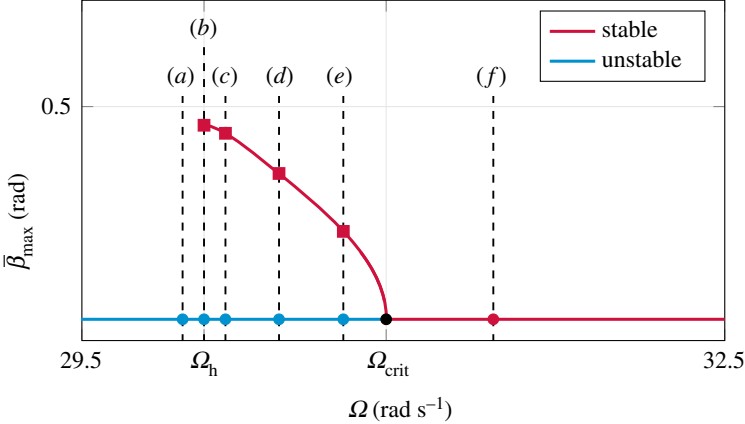

**Figure 8.** Location of the parameter values of figures 7 and 9 in the bifurcation diagram. Unstable inverted spinning is marked as blue dots, whereas stable spinning is indicated by a red dot. Stable periodic solutions are shown as red squared marks. The branch of periodic solutions has been obtained by numerical shooting. (Online version in colour.)

law has been assumed. Due to this linear friction law, the attractivity of the slow manifold $\mathcal{M}_s$ is global, i.e. would attract all solutions. However, here we have to bear in mind that we have made the approximation of a small smoothing parameter $\varepsilon$. For the chosen parameters, applying a smooth Coulomb friction (i.e. a nonlinear friction law) does not change the qualitative behaviour, since the slow manifold still seems to be globally attractive. However, this statement is based on numerical studies, as it is not trivial to prove. The critical manifold $\mathcal{M}_c$ defining pure rolling, approximates the slow manifold $\mathcal{M}_s$ to zero order. Since all solutions are attracted to the slow manifold $\mathcal{M}_s$ and thus also lie near the critical manifold, the relative sliding velocity must be small, justifying the assumed linearized version of smooth Coulomb friction (i.e. linear and smooth Coulomb friction describe the same asymptotic behaviour).

Due to the singularly perturbed structure, the long-term behaviour of the dynamics of the tippedisk is governed by a two-dimensional system describing the dynamics on the slow manifold $\mathcal{M}_s$. This slow manifold is approximated to zero order $\mathcal{O}(\varepsilon^0)$ by the critical manifold $\mathcal{M}_c$. This approximation suggests the reduction of the dynamics onto the critical manifold, like it is often assumed (e.g. [27]). Interestingly, this approximation is not sufficient to study the inversion phenomenon of the tippedisk. Since the critical manifold characterizes pure rolling, a reduction on the critical manifold is not able to capture the friction-induced instability of non-inverted spinning, nor the Hopf bifurcation at inverted spinning. Hence, the slow manifold must be approximated at least up to order $\mathcal{O}(\varepsilon)$, to study the behaviour 'near' pure rolling.

Figure 7 shows that the periodic solution defines an asymptotic attractive limit set embedded in the two-dimensional slow manifold. For $\Omega = \Omega_h$, the periodic solution degenerates into a heteroclinic cycle consisting of two heteroclinic connections on non-inverted spinning equilibria. Physically, both non-inverted equilibria can be identified with themselves, since both describe the same non-inverted spinning solution, and hence we may also speak of homoclinic connections. The birth of the stable periodic solution at $\Omega_h$ separates the slow manifold into two invariant sets, namely the 'interior', containing the inverted spinning solution, and the 'exterior'. If the spinning speed is subhomoclinic $\Omega < \Omega_h$, solutions are repelled from inverted spinning $\mathbf{x}_{in}$. For $\Omega_h < \Omega < \Omega_{crit}$, the inverted spinning equilibrium remains unstable and orbits starting near inverted and non-inverted spinning are attracted by the stable periodic solution. For increasing $\Omega$ the amplitude $\bar{\beta}_{max}$ and the period time $T$ are decreasing, until the supercritical Hopf bifurcation occurs at the bifurcation point $\Omega_{crit}$. After crossing this Hopf bifurcation, i.e. if the spinning speed $\Omega$ is higher than the critical spinning speed $\Omega_{crit}$ derived in [19], the inverted spinning solution attracts almost all trajectories, so that these orbits end up in an inverted configuration.

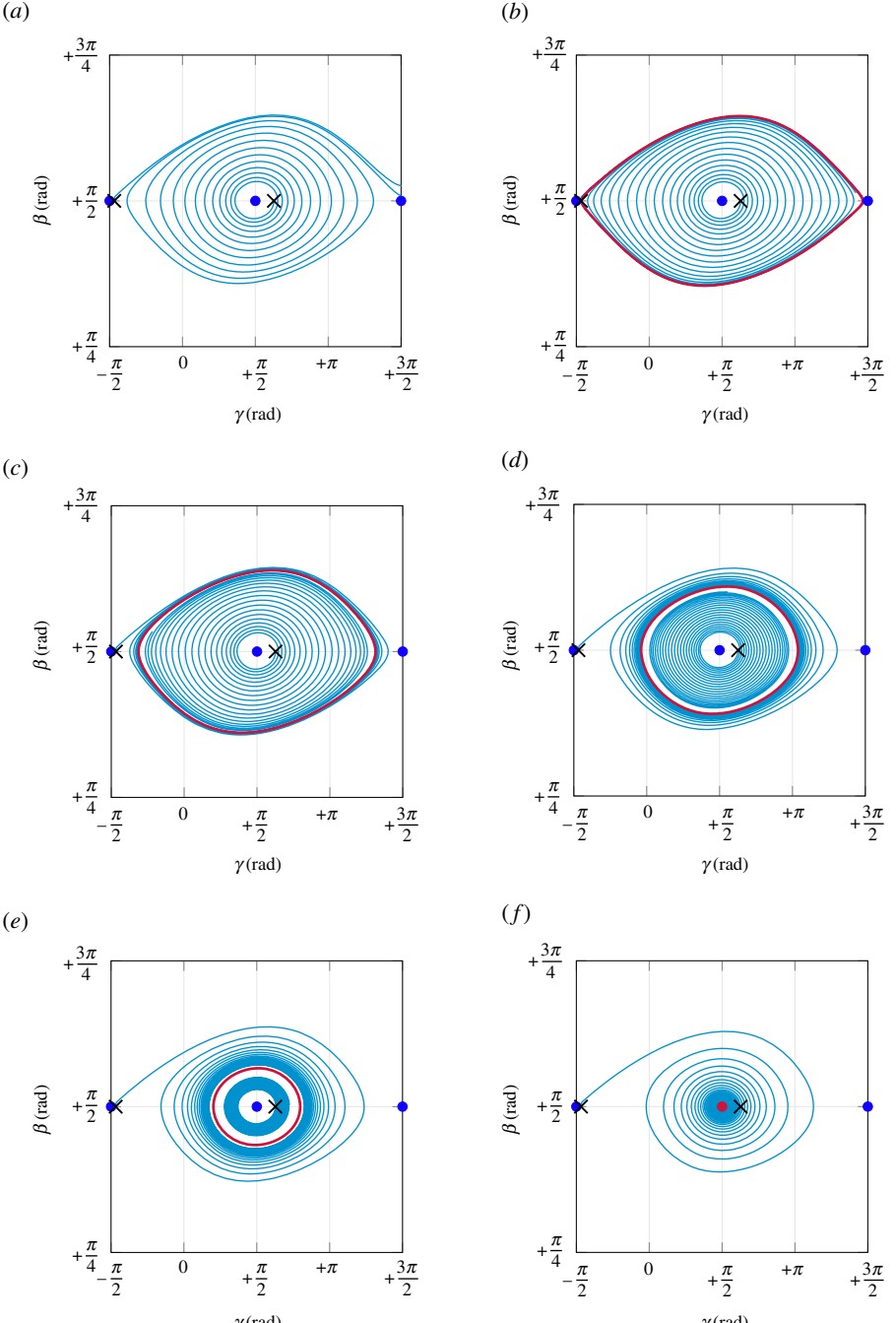

**Figure 9.** Projected three-dimensional dynamics onto $(\beta, \gamma)$-plane, corresponding to figure 7. (*a*) $\Omega = \Omega_{\mathrm{h}} - 0.1\,\mathrm{rad\ s^{-1}}$, (*b*) $\Omega = \Omega_{\mathrm{h}} + 0.01\,\mathrm{rad\ s^{-1}}$, (*c*) $\Omega = \Omega_{\mathrm{h}} + 0.1\,\mathrm{rad\ s^{-1}}$, (*d*) $\Omega = \Omega_{\mathrm{crit}} - 0.5\,\mathrm{rad\ s^{-1}}$, (*e*) $\Omega = \Omega_{\mathrm{crit}} - 0.2\,\mathrm{rad\ s^{-1}}$, (*f*) $\Omega = \Omega_{\mathrm{crit}} + 0.5\,\mathrm{rad\ s^{-1}}$. (Online version in colour.)

## 7. Conclusion

In this work, the nonlinear dynamics of the tippedisk has been studied. The starting point of the analysis is a three-dimensional dynamical system, derived in [19]. To characterize the Hopf bifurcation at $\Omega_{\mathrm{crit}}$, a harmonic balance approach is applied, indicating the existence of a

periodic solution for subcritical spinning velocities and thus characterizing the bifurcation as a supercritical Hopf bifurcation. For a feasible HBM in closed form, a local approximation of the system equations has been used, restricting the validity to the neighbourhood of the bifurcation point $\Omega_{\mathrm{crit}}$. The results obtained from the harmonic balance approach are validated by the application of a numerical shooting method and show the sudden birth of a periodic solution at $\Omega_{\mathrm{h}}$ (far away from the Hopf bifurcation) followed by a vanishing at the critical spinning velocity $\Omega_{\mathrm{crit}}$, derived in [19]. Due to the singularly perturbed structure of the system, solutions on a 'fast' time scale are attracted to a slow manifold almost immediately. After this transient 'jump' on the boundary layer, the orbits remain on this slow manifold $\mathcal{M}_s$, so that the asymptotic behaviour is characterized by the dynamics on this manifold. Since the dimension of the slow manifold is two, the three-dimensional dynamics can be reduced to a two-dimensional first-order ODE that qualitatively describes the inversion phenomenon of the tippedisk. The qualitative dynamics of the two-dimensional system will be compared with experiments in future research.

In summary, the bifurcation scenario is characterized by a homoclinic bifurcation in which a stable periodic orbit arises, followed by a supercritical Hopf bifurcation, after which the periodic solution has disappeared. If the spinning speed is supercritical (i.e. $\Omega > \Omega_{\mathrm{crit}}$, where a closed-form solution exists for $\Omega_{\mathrm{crit}}$), the inverted spinning solution attracts almost all trajectories, leading to the inversion of the tippedisk.

Data accessibility. All relevant data can be found at: https://figshare.com/s/d5bce39f401458a79f38 with corresponding DOI 10.6084/m9.figshare.16929508.

Authors' contributions. S.S., conceptualization: equal; data curation: lead; formal analysis: lead; investigation: lead; methodology: equal; software: lead; writing the original draft: lead. R.I.L., conceptualization: equal; methodology: equal; supervision: lead; writing the review: equal. Both authors approved the final version and agree to be accountable for all aspects of the work.

Competing interests. The authors declare that they have no conflicts of interest.

Funding. This research received no specific grant from any funding agency in the public, commercial or not-for-profit sectors.

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
