## [Peer Review File · Proceedings. Mathematical, Physical, and Engineering Sciences]

Review History

RSPA-2021-0536.R0 (Original submission)

Review form: Referee 1

Is the manuscript an original and important contribution to its field?

Excellent

Is the paper of sufficient general interest?

Excellent

Is the overall quality of the paper suitable?

Excellent

Can the paper be shortened without overall detriment to the main message?

Yes

Do you think some of the material would be more appropriate as an electronic appendix?

No

Do you have any ethical concerns with this paper?

No

Recommendation?

Accept with minor revision (please list in comments)

Comments to the Author(s)

The article is a continuation of the previous studies of the authors on the modeling and stability analysis of a so-called tippedisk, i.e. an eccentric disk that demonstrates dynamical properties resembling that of a tippe top or Jellet's egg that raise their centre of gravity when their spin exceeds some critical value. The system can easily be realized as a mechanical toy and therefore the mathematical model will be possible to compare with the upcoming experiments. The article is very well written, presenting the analysis and new results in full detail in a self-consistent, even pedagogical style. In addition to the linear stability analysis, providing an explicit expression for the critical spin, the authors apply a harmonic balance method to prove the supercritical character of the Hopf bifurcation and prove the existence of periodic solutions near the critical spin. Then, using the numerical shooting method in combination with continuation in the spin angular velocity and singular perturbation methods they identified the heteroclinic/homoclinic critical spinning velocity and illustrated the system's dynamics on a slow manifold. The exposition is excellent and I have only a few minor editorial comments.

1) in formulas (2) and (3) scalars are marked in bold and called matrix M and vector h , which is confusing. Perhaps, the authors' intention was just to keep the form of equations close to those that is used in the singular perturbation method later in the text. Perhaps, this could be stated explicitly on page 5?

2) Page 7 "convex conjugate" means probably "complex conjugate"?

3) Page 14 please check "yields the residual yields the four-dimensional residuum"

Perhaps, in the Introduction, the authors would be interested to mention recent experimental works on friction-induced instabilities, e.g. [D. Bigoni, O.N. Kirillov, D. Misseroni, G. Noselli, M. Tommasini (2018) Flutter and divergence instability in the Pfluger column: Experimental evidence of the Ziegler destabilization paradox.

Journal of the Mechanics and Physics of Solids, 116: 99-116] or another toy model that has many applications in Physics [O.N. Kirillov, M. Levi (2016) Rotating saddle trap as Foucault's pendulum. American Journal of Physics, 84(1): 26-31; O.N. Kirillov, M. Levi (2017) A Coriolis force in an inertial frame. Nonlinearity, 30(3): 1109-1119.]

Review form: Referee 2

Is the manuscript an original and important contribution to its field?

Good

Is the paper of sufficient general interest?

Good

Is the overall quality of the paper suitable?

Acceptable

Can the paper be shortened without overall detriment to the main message?

No

Do you think some of the material would be more appropriate as an electronic appendix?

No

Do you have any ethical concerns with this paper?

No

Recommendation?

Major revision is needed (please make suggestions in comments)

Comments to the Author(s)

Global analysis of dynamics of the tippedisk is carried out using the singular perturbation theory. It shows that the presence of friction leads to slow-fast dynamics and the creation of a two-dimensional slow manifold.

The research is interesting. However, major revisions are suggested as follows.

1. Novelty of the manuscript should be highlighted in the introduction, for it is not very clear what the real contribution of the paper is. Specially, what real engineering dynamics can be represented by the tippedisk? Dry-friction dynamics has been extensively investigated to reveal self-excited vibration phenomena. Is there new result obtained in this paper?
2. At beginning of section 4, it is mentioned that "the harmonic balance result is valid only for small amplitudes C and thus near the bifurcation point". In fact, high accuracy results for large amplitudes C can also be got by the harmonic balance method with high-order terms included in the expansions. Whereas the global dynamics is analyzed using the shooting method and singular perturbation theory in this section. If the main point of the investigation focuses on the methods, comparisons between the results obtained by different methods would be better provided. In addition, section 4.2.1 Basics of singular perturbation theory can be integrated with 4.2.2 for shortening the length of paper.

Review form: Referee 3

Is the manuscript an original and important contribution to its field?

Excellent

Is the paper of sufficient general interest?

Excellent

Is the overall quality of the paper suitable?

Excellent

Can the paper be shortened without overall detriment to the main message?

Yes

Do you think some of the material would be more appropriate as an electronic appendix?

Yes

Do you have any ethical concerns with this paper?

No

Recommendation?

Accept with minor revision (please list in comments)

Comments to the Author(s)

I thank the authors for their comments. I appreciated very much this publication and read it with great interest. My comments are of more general nature and rather minor see below

- p6 L 24: The Euler convention assumes rotation around three axes, you only mention zxz - do you mean zxx ?

- PRSA usually requests British English (<https://royalsociety.org/journals/authors/author-guidelines/>) - in different locations AE is used (linearized vs linearised etc).

- p6 L24: the explicit reference to equation $\alpha(t)$ as 'linear time evolution' is not correct= even though the time evolves linear, the function is an affine one ($f(t_1 + t_2) \neq f(t_1) + f(t_2)$); maybe just write "time evolution is expressed by the affine function..."
- In 3.1 it should be explicitly stated that the Hopf bifurcation point (non-linear system, time domain) does not fall together with the merging/splitting of the real/ imaginary parts. This is stated later on as far as I remember (when it comes to the nonlinear analysis) but I would recommend to pull this argument to the front. This would make explanation of linear stability more amenable and clear. Ω_1 is not explained in this context.
- now and then variables and results should be explained, eg in Eq 14 the significance of $O(1)$, $O(1/\epsilon)$ etc is not immediately obvious and should be explicitly mentioned (otherwise could be left away).
- make sure that all variables are explained - a nomenclature could be helpful (also as a control)
- please comment on the degree of simplification eg the neglect of higher order terms (p 9, L42) and discuss this in the conclusions of what could be expected.
- p13 L10, no comma "... point i.e., ..."
- Discussion section: It is not clear why Coloumb is referred to as 'linear friction law' (before) and then again as 'nonlinear and smooth friction law (Discussion)'. I assume it is referred to the linearised equation for the indirect Lyapunov method and the Coulomb friction for certain velocities as otherwise friction contains switching nonlinearities. Maybe go through the explanations and discussions and verify if this is clear.
- why is the attractively global if at least $O(\epsilon^0)$ are neglected? Please explain more here/ clarify (Discussion section)

Decision letter (RSPA-2021-0536.R0)

30-Sep-2021

Dear Mr Sailer

The Editor of Proceedings A has now received comments from referees on the above paper and would like you to revise it in accordance with their suggestions which can be found below (not including confidential reports to the Editor).

Please submit a copy of your revised paper within four weeks - if we do not hear from you within this time then it will be assumed that the paper has been withdrawn. In exceptional circumstances, extensions may be possible if agreed with the Editorial Office in advance.

Please note that it is the editorial policy of Proceedings A to offer authors one round of revision in which to address changes requested by referees. If the revisions are not considered satisfactory by the Editor, then the paper will be rejected, and not considered further for publication by the journal. In the event that the author chooses not to address a referee's comments, and no scientific justification is included in their cover letter for this omission, it is at the discretion of the Editor whether to continue considering the manuscript.

To revise your manuscript, log into <http://mc.manuscriptcentral.com/prsa> and enter your Author Centre, where you will find your manuscript title listed under "Manuscripts with Decisions." Under "Actions," click on "Create a Revision." Your manuscript number has been appended to denote a revision.

You will be unable to make your revisions on the originally submitted version of the manuscript. Instead, revise your manuscript and upload a new version through your Author Centre.

When submitting your revised manuscript, you will be able to respond to the comments made by the referee(s) and upload a file "Response to Referees" in Step 1: "View and Respond to Decision Letter". Please provide a point-by-point response to the comments raised by the reviewers and the editor(s). A thorough response to these points will help us to assess your revision quickly. You can also upload a 'tracked changes' version either as part of the 'Response to reviews' or as a 'Main document'.

IMPORTANT: Your original files are available to you when you upload your revised manuscript. Please delete any unnecessary previous files before uploading your revised version.

When revising your paper please ensure that it remains under 28 pages long. In addition, any pages over 20 will be subject to a charge (£150 + VAT (where applicable) per page). Your paper has been ESTIMATED to be 21 pages.

Open Access

You are invited to opt for open access, our author pays publishing model. Payment of open access fees will enable your article to be made freely available via the Royal Society website as soon as it is ready for publication. For more information about open access please visit <https://royalsociety.org/journals/authors/open-access/>. The open access fee for this journal is £1700/\$2380/€2040 per article. VAT will be charged where applicable. Please note that if the corresponding author is at an institution that is part of a Read and Publishing deal you are required to select this option. See <https://royalsociety.org/journals/librarians/purchasing/read-and-publish/read-publish-agreements/> for further details.

Once again, thank you for submitting your manuscript to Proc. R. Soc. A and I look forward to receiving your revision. If you have any questions at all, please do not hesitate to get in touch.

Yours sincerely
Raminder Shergill
proceedingsa@royalsociety.org

on behalf of
Professor Qu Shaoxing
Board Member
Proceedings A

Reviewer(s)' Comments to Author:

Referee: 1

Comments to the Author(s)

The article is a continuation of the previous studies of the authors on the modeling and stability analysis of a so-called tippedisk, i.e. an eccentric disk that demonstrates dynamical properties resembling that of a tippe top or Jellet's egg that raise their centre of gravity when their spin exceeds some critical value. The system can easily be realized as a mechanical toy and therefore the mathematical model will be possible to compare with the upcoming experiments. The article is very well written, presenting the analysis and new results in full detail in a self-consistent, even pedagogical style. In addition to the linear stability analysis, providing an explicit expression for the critical spin, the authors apply a harmonic balance method to prove the supercritical character of the Hopf bifurcation and prove the existence of periodic solutions near the critical spin. Then, using the numerical shooting method in combination with continuation in the spin angular velocity and singular perturbation methods they identified the heteroclinic/homoclinic critical spinning velocity and illustrated the system's dynamics on a slow manifold. The exposition is excellent and I have only a few minor editorial comments.

1) in formulas (2) and (3) scalars are marked in bold and called matrix M and vector h , which is confusing. Perhaps, the authors' intention was just to keep the form of equations close to those

that is used in the singular perturbation method later in the text. Perhaps, this could be stated explicitly on page 5?

2) Page 7 "convex conjugate" means probably "complex conjugate"?

3) Page 14 please check "yields the residual yields the four-dimensional residuum"

Perhaps, in the Introduction, the authors would be interested to mention recent experimental works on friction-induced instabilities, e.g. [D. Bigoni, O.N. Kirillov, D. Misseroni, G. Noselli, M. Tommasini (2018) Flutter and divergence instability in the Pfluger column: Experimental evidence of the Ziegler destabilization paradox.

Journal of the Mechanics and Physics of Solids, 116: 99-116] or another toy model that has many applications in Physics [O.N. Kirillov, M. Levi (2016) Rotating saddle trap as Foucault's pendulum. American Journal of Physics, 84(1): 26-31; O.N. Kirillov, M. Levi (2017) A Coriolis force in an inertial frame. Nonlinearity, 30(3): 1109-1119.]

Referee: 2

Comments to the Author(s)

Global analysis of dynamics of the tippedisk is carried out using the singular perturbation theory. It shows that the presence of friction leads to slow-fast dynamics and the creation of a two-dimensional slow manifold.

The research is interesting. However, major revisions are suggested as follows.

1. Novelty of the manuscript should be highlighted in the introduction, for it is not very clear what the real contribution of the paper is. Specially, what real engineering dynamics can be represented by the tippedisk? Dry-friction dynamics has been extensively investigated to reveal self-excited vibration phenomena. Is there new result obtained in this paper?
2. At beginning of section 4, it is mentioned that "the harmonic balance result is valid only for small amplitudes C and thus near the bifurcation point". In fact, high accuracy results for large amplitudes C can also be got by the harmonic balance method with high-order terms included in the expansions. Whereas the global dynamics is analyzed using the shooting method and singular perturbation theory in this section. If the main point of the investigation focuses on the methods, comparisons between the results obtained by different methods would be better provided. In addition, section 4.2.1 Basics of singular perturbation theory can be integrated with 4.2.2 for shortening the length of paper.

Referee: 3

Comments to the Author(s)

I thank the authors for their comments. I appreciated very much this publication and read it with great interest. My comments are of more general nature and rather minor see below

- p6 L 24: The Euler convention assumes rotation around three axes, you only mention zxz - do you mean zxz ?

- PRSA usually requests British English (<https://royalsociety.org/journals/authors/author-guidelines/>) - in different locations AE is used (linearized vs linearised etc).

- p6 L24: the explicit reference to equation $\alpha(t)$ as 'linear time evolution' is not correct= even though the time evolves linear, the function is an affine one ($f(t_1 + t_2) \neq f(t_1) + f(t_2)$); maybe just write "time evolution is expressed by the affine function..."

- In 3.1 it should be explicitly stated that the Hopf bifurcation point (non-linear system, time domain) does not fall together with the merging/splitting of the real/ imaginary parts. This is stated later on as far as I remember (when it comes to the nonlinear analysis) but I would recommend to pull this argument to the front. This would make explanation of linear stability more amenable and clear. Ω_1 is not explained in this context.

- now and then variables and results should be explained, eg in Eq 14 the significance of $O(1)$, $O(1/\epsilon)$ etc is not immediately obvious and should be explicitly mentioned (otherwise could be left away).
- make sure that all variables are explained - a nomenclature could be helpful (also as a control)
- please comment on the degree of simplification eg the neglect of higher order terms (p 9, L42) and discuss this in the conclusions of what could be expected.
- p13 L10, no comma "... point i.e., ..."
- Discussion section: It is not clear why Coloumb is referred to as 'linear friction law' (before) and then again as 'nonlinear and smooth friction law (Discussion). I assume it is referred to the linearised equation for the indirect Lyapunov method and the Coulomb friction for certain velocities as otherwise friction contains switching nonlinearities. Maybe go through the explanations and discussions and verify if this is clear.
- why is the attractively global if at least $O(\epsilon^0)$ are neglected? Please explain more here/clarify (Discussion section)

Author's Response to Decision Letter for (RSPA-2021-0536.R0)

See Appendix A.

RSPA-2021-0536.R1 (Revision)

Review form: Referee 1

Is the manuscript an original and important contribution to its field?

Excellent

Is the paper of sufficient general interest?

Excellent

Is the overall quality of the paper suitable?

Excellent

Can the paper be shortened without overall detriment to the main message?

Yes

Do you think some of the material would be more appropriate as an electronic appendix?

No

Do you have any ethical concerns with this paper?

No

Recommendation?

Accept as is

Comments to the Author(s)

I am fully satisfied with the revised version.

Review form: Referee 2

Is the manuscript an original and important contribution to its field?

Excellent

Is the paper of sufficient general interest?

Good

Is the overall quality of the paper suitable?

Good

Can the paper be shortened without overall detriment to the main message?

Yes

Do you think some of the material would be more appropriate as an electronic appendix?

No

Do you have any ethical concerns with this paper?

No

Recommendation?

Accept as is

Comments to the Author(s)

The manuscript has been well improved. I have no any suggestion for further improvement.

Review form: Referee 3

Is the manuscript an original and important contribution to its field?

Good

Is the paper of sufficient general interest?

Acceptable

Is the overall quality of the paper suitable?

Good

Do you have any ethical concerns with this paper?

No

Recommendation?

Accept as is

Comments to the Author(s)

I am satisfied with the author's comments and congratulate them to their work.

Decision letter (RSPA-2021-0536.R1)

02-Nov-2021

Dear Mr Sailer

I am pleased to inform you that your manuscript entitled "Singularly perturbed dynamics of the tippedisk" has been accepted in its final form for publication in Proceedings A.

Our Production Office will be in contact with you in due course. You can expect to receive a proof of your article soon. Please contact the office to let us know if you are likely to be away from e-mail in the near future. If you do not notify us and comments are not received within 5 days of sending the proof, we may publish the paper as it stands.

As a reminder, you have provided the following 'Data accessibility statement' (if applicable). Please remember to make any data sets live prior to publication, and update any links as needed when you receive a proof to check. It is good practice to also add data sets to your reference list.
Statement (if applicable):

Open access

You are invited to opt for open access, our author pays publishing model. Payment of open access fees will enable your article to be made freely available via the Royal Society website as soon as it is ready for publication. For more information about open access please visit <https://royalsociety.org/journals/authors/which-journal/open-access/>. The open access fee for this journal is £1700/\$2380/€2040 per article. VAT will be charged where applicable.

Note that if you have opted for open access then payment will be required before the article is published – payment instructions will follow shortly.

If you wish to opt for open access then please inform the editorial office (proceedingsa@royalsociety.org) as soon as possible.

Your article has been estimated as being 21 pages long. Our Production Office will inform you of the exact length at the proof stage.

Proceedings A levies charges for articles which exceed 20 printed pages. (based upon approximately 540 words or 2 figures per page). Articles exceeding this limit will incur page charges of £150 per page or part page, plus VAT (where applicable).

Under the terms of our licence to publish you may post the author generated postprint (ie. your accepted version not the final typeset version) of your manuscript at any time and this can be made freely available. Postprints can be deposited on a personal or institutional website, or a recognised server/repository. Please note however, that the reporting of postprints is subject to a media embargo, and that the status the manuscript should be made clear. Upon publication of the definitive version on the publisher's site, full details and a link should be added.

You can cite the article in advance of publication using its DOI. The DOI will take the form: 10.1098/rspa.XXXX.YYYY, where XXXX and YYYY are the last 8 digits of your manuscript number (eg. if your manuscript number is RSPA-2017-1234 the DOI would be 10.1098/rspa.2017.1234).

For tips on promoting your accepted paper see our blog post:

<https://royalsociety.org/blog/2020/07/promoting-your-latest-paper-and-tracking-your-results/>

On behalf of the Editor of Proceedings A, we look forward to your continued contributions to the Journal.

Sincerely,
Raminder Shergill
proceedingsa@royalsociety.org

on behalf of
Professor Qu Shaoxing
Board Member
Proceedings A

Reviewer(s)' Comments to Author:
Referee: 1
Comments to the Author(s)
I am fully satisfied with the revised version.

Referee: 3
Comments to the Author(s)
I am satisfied with the author's comments and congratulate them to their work.

Referee: 2
Comments to the Author(s)
The manuscript has been well improved. I have no any suggestion for further improvement.

Appendix A

Answers to the reviewers

We thank the reviewers for their useful comments. All comments have been taken into account in the final manuscript. We think that due to these comments the paper has significantly improved. Changes in our manuscript are highlighted in blue.

- **Referee 1:**

“The article is a continuation of the previous studies of the authors on the modeling and stability analysis of a so-called tippedisk, i.e. an eccentric disk that demonstrates dynamical properties resembling that of a tippe top or Jellet’s egg that raise their centre of gravity when their spin exceeds some critical value. The system can easily be realized as a mechanical toy and therefore the mathematical model will be possible to compare with the upcoming experiments. The article is very well written, presenting the analysis and new results in full detail in a self-consistent, even pedagogical style. In addition to the linear stability analysis, providing an explicit expression for the critical spin, the authors apply a harmonic balance method to prove the supercritical character of the Hopf bifurcation and prove the existence of periodic solutions near the critical spin. Then, using the numerical shooting method in combination with continuation in the spin angular velocity and singular perturbation methods they identified the heteroclinic/homoclinic critical spinning velocity and illustrated the system’s dynamics on a slow manifold. The exposition is excellent and I have only a few minor editorial comments.”

We thank the reviewer for his appraisal.

Answers to comments of Referee 1:

1. “in formulas (2) and (3) scalars are marked in bold and called matrix M and vector h , which is confusing. Perhaps, the authors’ intention was just to keep the form of equations close to those that is used in the singular perturbation method later in the text. Perhaps, this could be stated explicitly on page 5?”
The scalar mass ‘matrix’ \$M\$ and the scalar ‘vector’ \$h\$ are marked in bold for consistency with previous works, since the initial system yields a mechanical structure of the general form \$M\ddot{\mathbf{q}} - \mathbf{h}(\mathbf{q}, \dot{\mathbf{q}}) = \mathbf{f}(\mathbf{q}, \dot{\mathbf{q}})\$ in Lagrangian coordinates \$\mathbf{q} \in \mathbb{R}^n\$. Moreover, this form allows the introduction of the singular perturbation method from a general point of view, i.e., valid also for higher dimensional mechanical systems.
To clarify this, we have made a footnote in the text on page 6.
2. “Page 7 “convex conjugate” means probably “complex conjugate”?”
You are right, of course.
This error has been corrected.
3. “Page 14 please check “yields the residual yields the four-dimensional residuum””
We changed this on page 14.
4. “Perhaps, in the Introduction, the authors would be interested to mention recent experimental works on friction-induced instabilities, e.g. [D. Bigoni,

O.N. Kirillov, D. Misseroni, G. Noselli, M. Tommasini (2018) Flutter and divergence instability in the Pfluger column: Experimental evidence of the Ziegler destabilization paradox. *Journal of the Mechanics and Physics of Solids*, 116: 99-116] or another toy model that has many applications in Physics [O.N. Kirillov, M. Levi (2016) Rotating saddle trap as Foucault's pendulum. *American Journal of Physics*, 84(1): 26-31; O.N. Kirillov, M. Levi (2017) A Coriolis force in an inertial frame. *Nonlinearity*, 30(3): 1109-1119.]”
The manuscript focuses on friction-induced instability phenomena of single rigid bodies, such as Euler's disk or the tippetop. Since we aim to keep this focus, we refrain from presenting additional scientific toys, of which there are an incredible number, so as not to steer the reader in the wrong direction.

- **Referee 2:**

“Global analysis of dynamics of the tippedisk is carried out using the singular perturbation theory. It shows that the presence of friction leads to slow-fast dynamics and the creation of a two-dimensional slow manifold. The research is interesting. However, major revisions are suggested as follows.”

We thank the reviewer for his useful comments.

Answers to comments of Referee 2:

1. “Novelty of the manuscript should be highlighted in the introduction, for it is not very clear what the real contribution of the paper is. Specially, what real engineering dynamics can be represented by the tippedisk? Dry-friction dynamics has been extensively investigated to reveal self-excited vibration phenomena. Is there new result obtained in this paper?”

In the scientific research community there are many academic systems, e.g., the tippetop, the rattleback, Euler's disk, showing fascinating nonlinear behavior. As these system can easily be realized as mechanical toys, it is possible to compare mathematical models with experiments. Even though the systems mentioned are originally from the last century and have little to do with engineering applications, they have a huge impact in the scientific community and are part of current research. The tippedisk forms a new scientific toy showing an interesting inversion phenomenon and serves as a mechanical mathematical archetype for friction-induced instability phenomena. The nonlinear analysis presented in this manuscript describes for the first time the qualitative behavior of the mechanical system ‘tippedisk’ on a mathematical level and shows the application of singular perturbation theory to this mechanical system, leading to global bifurcations. Therefore, this manuscript sets the starting point for further in depth research. Due to the structure of the system equations and the associated friction laws, singular perturbation theory can be applied to vehicle dynamics models in a manner analogous to the tippedisk, which is thus an engineering application. This has been highlighted in the introduction.

2. “At beginning of section 4, it is mentioned that “the harmonic balance result is valid only for small amplitudes C and thus near the bifurcation point”. In fact, high accuracy results for large amplitudes C can also be got by the harmonic balance method with high-order terms included in the expansions. Whereas the global dynamics is analysed using the shooting method and singular perturbation theory in this section. If the main point of the investigation focuses on the methods, comparisons between the results obtained by different methods would be better provided. In addition, section 4.2.1 Basics of singular perturbation theory can be integrated with 4.2.2 for shortening the length of paper.”

Of course, the multi harmonic balance method, taking into account higher order terms, gives accurate results even for large amplitudes C . The aim of Section 3 is to find a closed-form expression which characterizes the Hopf bifurcation. To obtain closed-form expressions, the single harmonic balance method is applied, neglecting higher order terms, which results in an approximate solution being valid only around the equilibrium. This has been already stated on Page 14 of the submitted manuscript: "The search for closed-form expressions necessitates local approximations of the dynamics by neglecting higher order terms. For this reason, the harmonic balance result is valid only for small amplitudes C and thus near the bifurcation point." A general comparison of the shooting method and the harmonic balance is not the aim of this manuscript. Numerical shooting serves on the one hand to check the closed-form results and on the other hand to search globally for periodic solutions. Alternatively, one could have used the multi harmonic balance method numerically, which would have led to the same results. To make this clearer, we have added additional comments in the revised manuscript.

To keep the paper accessible to both mathematicians and engineers, we prefer to introduce the singular perturbation theory in general before applying the analysis to the tippedisk.

- **Referee 3:**

“I thank the authors for their comments. I appreciated very much this publication and read it with great interest. My comments are of more general nature and rather minor see below”

The authors thank the referee for his accurate reading of the paper.

Answers to comments of Referee 3:

1. “ p6 L 24: The Euler convention assumes rotation around three axes, you only mention zxz - do you mean zxx' ?”

Thank you for this hint. We use an intrinsic parametrisation and therefore the zxz sequence defines a rotation around the subsequently ‘rotated’ coordinate frame, i.e., first a rotation in the inertial frame, second a rotation in the rotated frame and third the rotation in the twice-rotated frame. I assume that you use $'$ to express a rotation around a rotated axis. In this case our

chosen parametrisation may be expressed as $zx'z$ sequence. For consistency with previous work that discuss the parametrisation in detail, we prefer the notation zxx , but have made this clearer in the text. Therefore, the following text has been added on page 5: “It is worth mentioning here that the angles correspond to an intrinsic parametrisation, not to be confused with an extrinsic description.”

2. “ PRSA usually requests British English (<https://royalsociety.org/journals/authors/author-guidelines/>) - in different locations AE is used (linearized vs linearised etc).”

Thank you very much for pointing this out. This has been corrected.

3. “p6 L24: the explicit reference to equation $\alpha(t)$ as 'linear time evolution' is not correct= even though the time evolves linear, the function is an affine one ($f(t1 + t2) \neq f(t1) + f(t2)$); maybe just write "time evolution is expressed by the affine function...””

We assume you mean L32 on page 6. We made this clearer.

4. “In 3.1 it should be explicitly stated that the Hopf bifurcation point (non-linear system, time domain) does not fall together with the merging/splitting of the real/ imaginary parts. This is stated later on as far as I remember (when it comes to the nonlinear analysis) but I would recommend to pull this argument to the front. This would make explanation of linear stability more amenable and clear. Ω_1 is not explained in this context.”

Your are right. The spinning velocity Ω_1 at the splitting point is not introduced properly, because it is a relic from a previous paper [1]. Since Ω_1 does not play an important role in this manuscript, we have removed the axis label in Figure 4 to avoid confusing the reader. Moreover, the caption of Figure 4 has been adapted for clarity.

5. “now and then variables and results should be explained, eg in Eq 14 the significance of $O(1)$, $O(1/\epsilon)$ etc is not immediately obvious and should be explicitly mentioned (otherwise could be left away).”

For the following closed-form analysis, the orders \mathcal{O} of the matrix coefficients A_{ij} play an important role. To explain this better in the text, we added on page 8: “As we see from Eq. (14), the matrix coefficient A_{31} does not depend on the smoothing parameter ϵ and is therefore of order $\mathcal{O}(1)$. Both A_{32} and A_{33} depend proportional on $\frac{1}{\epsilon}$ and are therefore of order $\mathcal{O}(\frac{1}{\epsilon})$.”

6. “make sure that all variables are explained - a nomenclature could be helpful (also as a control)”

We refrain to introduce a nomenclature, but have verified that the variables are explained in the text. If this was not the case, changes were made in the text.

7. “please comment on the degree of simplification eg the neglect of higher order terms (p 9, L42) and discuss this in the conclusions of what could be

expected.”

To classify the nature of the Hopf bifurcation correctly, the local approximation Eq. (21) must contain all cubic terms. This can be achieved by approximating the trigonometric expressions Eqs. (17-20) up to the first nonlinear part. We made a comment on page 9 in the revised manuscript.

8. “p13 L10, no comma "... point i.e., ...”

We are not sure if you mean the comma after ‘i.e.’. If so, this is a relic of American English.

9. “Discussion section: It is not clear why Coloumb is referred to as ‘linear friction law’ (before) and then again as ‘nonlinear and smooth friction law’ (Discussion). I assume it is referred to the linearised equation for the indirect Lyapunov method and the Coulomb friction for certain velocities as otherwise friction contains switching nonlinearities. Maybe go through the explanations and discussions and verify if this is clear.”

The classical Coulomb friction law is indeed nonlinear and even set-valued. In [1] it is shown that the regularisation of the Coulomb friction law is sufficient to describe the inversion of the tippedisk. This regularisation, also known as ‘smooth Coulomb friction law’, can be linearised around zero slip, which is referred to as ‘linear Coulomb friction’. Near the slow manifold, the relative sliding velocity is close to zero, so both regularised/smooth and linear Coulomb friction lead to the same qualitative dynamical behaviour. We have made comments on page 6 and in the discussion section.

10. “why is the attractively global if at least $O(\epsilon^0)$ are neglected? Please explain more here/ clarify (Discussion section)”

The attraction of the slow manifold is studied by a linearised distance dynamics on the fast timescale. For linear Coulomb friction it can be proved that the slow manifold, being an equilibrium on the boundary layer, is globally attractive. The smoothing coefficient ε is small and has to be understood as parameter and does not define the local neighbourhood in which the linearisation holds. We made this clearer in the text.

References

- [1] S. Sailer and R. I. Leine. Model reduction of the tippedisk: a path to the full analysis. *Nonlinear Dyn*, 105(3):1955–1975, 2021.